# Ammonia Synthesis over Transition Metal Catalysts: Reaction Mechanisms, Rate-Determining Steps, and Challenges

**DOI:** 10.3390/ijms26104670

**Published:** 2025-05-13

**Authors:** Pradeep R. Varadwaj, Helder M. Marques, Ireneusz Grabowski

**Affiliations:** 1Molecular Sciences Institute, School of Chemistry, University of the Witwatersrand, Johannesburg 2050, South Africa; 2Institute of Physics, Faculty of Physics, Astronomy, and Informatics, Nicolaus Copernicus University in Toruń, 87-100 Toruń, Poland

**Keywords:** ammonia synthesis, Haber–Bosch process, concepts and applications, elementary reaction mechanisms, reaction order, rate-determining step

## Abstract

Ammonia synthesis remains a cornerstone of global chemical manufacturing, essential for fertilizer production, energy storage, and emerging carbon capture technologies. This overview examines recent developments in the understanding of elementary reaction mechanisms in heterogeneous catalysis, with emphasis on transition metal thermocatalysts operating under the Haber–Bosch process. Traditionally, the dissociative adsorption of nitrogen (N_2_) has been considered the rate-determining step. However, recent studies challenge this view, revealing possible shifts in rate-determining steps and suggesting that alternative mechanistic pathways may be operative. The discussion critiques studies that adhere strictly to the classic dissociative mechanism—often inferred from the reaction order of N_2_—while overlooking alternative pathways that could offer more efficient catalytic routes and deeper mechanistic insight into ammonia synthesis. These insights offer a pathway toward more rational catalyst design and improved process efficiency in ammonia synthesis.

## 1. Introduction

The energy-intensive Haber–Bosch process [1], an unparalleled breakthrough developed by the brilliant minds of Fritz Haber and Carl Bosch [2,3,4]—both of whom were honored with Nobel Prizes in Chemistry (1918 and 1931, respectively)—has stood as the bedrock of industrial ammonia synthesis for over a century [5], revolutionizing the large-scale production of ammonia, which is indispensable to the global agricultural system [1,6,7,8].

The intellectual contributions of countless pioneers have enriched the legacy of this monumental achievement [9], including Emmett et al. [10,11] for their work on adsorption and catalyst design, particularly with iron-based catalysts; Mittasch et al. [8,12] for their exploration of osmium-, uranium-, and fused Fe-based catalysts; and Ertl et al. [13,14] for their groundbreaking contributions to understanding surface reactions at the atomic level.

Aika and Ozaki et al. [15,16] expanded the horizons of ruthenium and other transition metal catalysts, opening up the possibility of more efficient processes under milder conditions than iron-based catalysts. Somorjai et al. [17,18] added to our fundamental understanding of the role of promoters in iron-and ruthenium-based catalysts. Nørskov et al. [19,20,21] made significant conceptual contributions to heterogeneous catalysis, particularly concerning the volcano plot and the Brønsted–Evans–Polanyi relation that links kinetics (activation barriers) with thermodynamics (reaction enthalpies) [22,23]. Hosono et al. [24,25,26,27] have continued to inspire the field through their exploration of transition metal nitride-, hydride-, and electride-based catalysts, aiming to improve the energy efficiency of the process and reduce the harsh conditions required in traditional methods for ammonia synthesis. Taylor’s theory of catalytic surfaces [28], Boudart’s work on the paradox of heterogeneous kinetics [29,30,31], Temkin and Pyzhev’s [31] adsorption isotherm theory—subsequently applied to the performance of a Haber–Bosch reactor across a wide range of operating conditions by Krewer et al. [32]—which emphasizes nitrogen adsorption on the catalyst surface, and its correction for diffusion by Dyson and Simon [33] and Stoltze et al. [34,35], have all significantly advanced the field. Spencer’s work on chemical kinetics and mixed alloy catalysts [36,37,38], Bowker’s development of rate equations assuming dissociative hydrogen adsorption in two steps, along with several hydrogenation steps [39], and Brill’s concept of poisoning in ammonia synthesis [40] have further shaped our understanding of the ammonia synthesis activity process.

The reaction between nitrogen and hydrogen gases in a 1:3 ratio is exothermic [41,42]. N_2_ (g) + 3H_2_ (g) ⇌ 2NH_3_ (g), ΔH = −92 kJ/mol(1)

The Haber–Bosch process for ammonia synthesis operates under extreme conditions, with temperatures of 375–500 °C and pressures of 200–300 atm [43]. These conditions are necessary to overcome the high activation energy barrier of N_2_ bond dissociation (941.4 kJ mol^−1^ [44]), but they also demand substantial energy input (~485 kJ mol^−1^). Despite its significance, the process remains challenged by energy efficiency and catalytic performance limitations. Consequently, ongoing innovation aims to enhance sustainability and efficiency for future generations [41]. At the heart of these advancements is the pursuit of novel catalysts capable of operating at significantly lower temperatures and pressures, reducing the need for energy-intensive compression and ushering in a new era of energy-efficient ammonia synthesis.

For decades, it was unclear whether adsorbed hydrogen atoms interacted with nitrogen in its atomic or molecular form on the catalyst surface. An Auger spectroscopy study [45,46] resolved this uncertainty by revealing that the density of adsorbed atomic nitrogen sharply decreases at elevated hydrogen pressures. This insight enhances our understanding of the adsorption process: at low pressure, hydrogen and nitrogen are weakly to moderately adsorbed onto the catalyst surface [9,11]. As both pressure and temperature increase, these molecules undergo dissociative adsorption, with hydrogen dissociating more rapidly than nitrogen. This differential behavior is pivotal for the subsequent reaction step.

Central to the Haber–Bosch process is the use of a metallic catalyst to drive reaction (1). Typically, the metal starts in a mixed-oxidation state—often as iron oxides like Fe_3_O_4_. In the initial step, gaseous N_2_ and H_2_ molecules adsorb onto the catalyst surface even if it is not fully reduced. The efficiency of N≡N bond cleavage, however, depends critically on the catalyst’s reduction state.

Once the catalyst is fully reduced to its metallic form (Fe^0^ [10,47] and Ru^0^ [48]), it facilitates the thermodynamically favorable chemisorption of two N atoms onto two separate sites (N_2_ → 2N) [31]. These reactive sites are limited by the number of vacant adsorption sites available on the surface [28]. If the catalyst remains poorly reduced, the chemisorbed N_2_ first reacts with adsorbed hydrogen to form a less stable N–NH (or HN–NH) intermediate, which then dissociates into adsorbed NH radical intermediates. This sequential reduction is essential for effective interactions between the adsorbed nitrogen and hydrogen atoms, ultimately leading to ammonia formation [8,49,50].

Moreover, the catalyst’s microstructure—comprising metal particles interspersed with unreduced oxides or nitrides—promotes the formation of metal–nitrogen intermediates [51], thereby enhancing catalytic efficiency.

A crucial aspect of the Haber–Bosch process is identifying the rate-determining step (RDS) [24,26,32,52,53]—the slowest step that governs the overall reaction rate on transition metal catalysts [49,50], such as Fe and Fe_2_O_4_ [54,55]. Traditionally, this step was attributed to the dissociative adsorption of N_2_ [25,32,42,56,57], a concept first proposed by Emmett and Brunauer [11]. However, recent studies highlight that the RDS can vary depending on reaction conditions, catalyst composition, and the influence of structural [58] and electronic promoters [52,59,60,61,62]. Advances in catalyst microstructural analysis and promoter effects have refined our understanding of the high-temperature dissociative pathway [63], while also pioneering alternative reaction mechanisms. These insights enable more efficient ammonia synthesis under milder conditions [64] by leveraging associative, alternating, or distal enzymatic hydrogenation mechanisms, ultimately enhancing control over the hydrogenation process [65].

Transition metal catalysts—such as Fe, Co, Mo, and Ru [66,67,68,69]—alone are often insufficient for efficient ammonia synthesis. To enhance their performance, they are typically paired with either structural or electronic promoters. Electronic promoters, including alkali, alkaline earth, or lanthanide metals (or their oxides and hydrides), improve catalytic activity by modifying the electronic environment. For example, Fe-based catalysts combined with Al_2_O_3_ and K_2_O exhibit enhanced N_2_ triple bond cleavage and reduced bond dissociation energy [12,61,70,71]. Al_2_O_3_, a structural promoter, prevents sintering by spacing small iron platelets and may also stabilize Fe(111) facets, optimizing catalytic performance. By contrast, K_2_O functions as an electronic promoter by facilitating charge transfer to the transition metal surface, altering reaction orders for ammonia and hydrogen [72], and lowering the activation energy for N_2_ dissociation—ultimately increasing ammonia synthesis rates [58,73]. Strongly basic supports such as CeO_2_, La_0.5_Pr_0.5_O_1.75_, Ba/Ce_0.5_La_0.5_O_1.75_, CeO_2_-PrO_x_, and Ce_0.5_La_0.5−x_Ti_x_O_1.75+0.5x_, contribute to the promotion of N_2_ dissociation. They enhance the transfer of electron density from Ru to the antibonding orbital of N_2_, facilitating the dissociation process [74].

Barium, as an electronic promoter, is believed to modify surface morphology by forming highly active B5-type sites on Ru catalysts, thereby enhancing catalytic efficiency [75]. In the case of a Ru/Sr_2_Ta_2_O_7_ catalyst, ammonia synthesis activity correlates with the electron donation capacity of alkali and alkaline-earth promoters, Cs > Rb > K > Ba [76]. Some studies further identify Cs as the strongest electron donor, while BaO primarily functions as a hydrogen scavenger [77]. The promoting mechanism and activity can vary depending on metal-oxide combinations, with hydroxide promoters exhibiting effectiveness in the order Cs_2_O > K_2_O > Na_2_O > BaO > CaO [78]. Under specific conditions, these promoters can shift the rate-limiting step from N_2_ dissociation to hydrogenation processes, such as the hydrogenation of nitrogen intermediates or ammonia desorption. In the case of the promoters K, Cs, Ba, and Li, the reactivity order for promoted Ru/La_2_Ce_2_O_2_-suppoted catalysts has been reported as K > Cs > Ba > Li (at 400 °C and 1 MPa), which contradicts the order based on the electronegativity of the promoters (Li > Ba > K > Cs) [48]. Furthermore, it has been shown that anchoring Ba and/or Ce onto Ru atomic cluster catalysts (ACCs) significantly enhances the catalytic N_2_-to-NH_3_ conversion, boosting the rate of NH_3_ synthesis to 56.2 mmol NH_3_ g_at_^−1^ h^−1^ at 400 °C and 1 MPa—7.5 times higher than that of Ru ACC alone [79].

This overview summarizes studies investigating how the rate-determining step changes across various catalysts—including Fe, Co, Ru, and transition metal alloys—and how these variations influence activation energy [59]. We begin by analyzing the activation sites on selected crystal phases of these metals, focusing on both molecular and dissociative adsorption states, as well as activation profiles and energies, typically characterized using experimental techniques and/or theoretical approaches such as density functional theory (DFT) [80,81]. The discussion highlights key catalyst systems where specific elementary reaction mechanisms contribute to a deeper understanding of N_2_ decomposition in ammonia synthesis. Additionally, we explore the effects of promoter incorporation on catalytic performance, wherever applicable, and assess recent progress in establishing a qualitative relationship between reaction order and the RDS. This overview provides a concise perspective on the research required to advance catalytic design, improve efficiency, and enhance the sustainability of ammonia synthesis.

## 2. Catalytic Surface Reactivity Through the Lens of Geometry: Adsorption Site Topologies and Their Role in Heterogeneous Catalysis

In the microscopic realm of catalysis, it’s not pristine perfection but structural irregularity that gives rise to exceptional reactivity, as discussed by Taylor [28], whose view ultimately proved more accurate than Langmuir’s earlier perspective [82]. Catalysts operate most effectively when their surfaces are not flat and featureless but instead exhibit irregular contours—ridges, ledges, and atomic-scale discontinuities (defects). These imperfections serve as active zones that can transiently capture and orient reactant molecules, enhancing their likelihood of undergoing transformation. On such surfaces, molecules are not evenly spread or randomly placed—they gravitate toward specific topological features where the (adsorption) binding energy is most favorable.

Ruthenium commonly exhibits two principal surface structures: flat terraces [83,84] and step edges [85,86,87]—features that may also be characteristic of cobalt surfaces. Typical adsorption sites on the flat surface include on-top (t), bridge (b), face-centered cubic (fcc/FCC), hexagonal close-packed (hcp/HCP), as illustrated in Figure 1a,b. Wulff constructions of hcp metals such as Ru and Co nanoparticles typically feature dominant facets including the basal plane (0001), the prismatic plane (10$\bar{1}$0) and the vicinal stepped surface (10$\bar{1}$1). The latter corresponds to a vicinal plane inclined from the basal plane and features regular monatomic-height steps. These facets are energetically favorable and are commonly observed in both experimental and theoretical studies of hcp metal nanoparticles and single crystals [88].

The B5 adsorption site [89,90] is of particular catalytic significance. Found predominantly on Ru(0001) and Co(0001) step edges, the B5 site comprises five metal atoms: two at the step edge and three on the adjoining terrace, arranged in a distinctive configuration. This geometry provides an energetically favorable environment for adsorption, enabling the site to stabilize reactant species just long enough to facilitate key chemical transformations—thereby making it highly active in catalytic processes. Figure 1b,c illustrate a stepped surface of hcp Ru featuring the B5 site that consists of five Ru atoms arranged to expose both a three-fold hollow (hcp) site and a nearby bridge site in close proximity, analogous to that reported elsewhere [91].

The fcc(211) and bcc(310) surfaces are prototypical stepped terminations that inherently expose B5-active sites due to their distinct atomic arrangements. By contrast, flat surfaces such as hcp(0001) or fcc(111) lack the necessary step or kink features to host B5 sites. However, fcc(111), while atomically smooth, may be used to construct vicinal surfaces like fcc(211) through directional cleaving, thereby introducing step edges where B5 geometries can emerge. Similarly, if an Ru(0001) slab is constructed from a cubic lattice or any non-hexagonal surface, B5 sites potentially can still form, provided that step edges are introduced and the local atomic arrangement supports the specific fivefold geometry required for a B5 site (Figure 1b). The hexagonal Co(10$\bar{1}$5) and Ru(10$\bar{1}$5) facets have been reported to display step structures [92], a member of the (10$\bar{1}$*l*) (l=1-9) family. These surfaces are vicinal to the (0001) plane, with terraces aligned along the [12$\bar{1}$0] direction and increasing step height as *l* increases. The Ru(11$\bar{2}$*l*) (*l* = 1-3) surfaces may have slightly different terrace orientations and compared to Ru(10$\bar{1}$5), with higher densities of steps or kinks, and could display more complex step-edge geometries due to the increasing step height and terrace misalignment. 

The B6 adsorption site—consisting of a six-atom cluster made up of two step-edge atoms and four terrace atoms—has been identified on Ru catalysts but remains relatively underexplored as a potentially active adsorption site [90]. Structurally, it can be viewed as a distorted octahedral cavity, as may be inferred from Figure 1c. B-active sites, which are not always readily observed on small nanoparticles (NPs), become more apparent on large-sized, highly reactive NPs and can be tracked using Wulff construction in combination with first-principles calculations [90].

Zeinalipour-Yazdi’s study investigated the diversity of local active site geometries on metal NPs and the surfaces of FCC and HCP metals [93]. As many as 18 local site geometries were characterized in HCP and FCC structures. These adsorption sites can be described by combinations of equilateral triangles and squares, with atoms adopting either bridge or atop bonding conformations. The geometries are further distinguished by their alignment above tetrahedral or octahedral hollows. Notably, the study proposed that the well-known B5 site (equilateral-triangle-square with a common side)—previously identified as catalytically active in ammonia synthesis and decomposition—may exist in five structural variants, depending on the angle between the triangular and square motifs. While this classification had not been explicitly formalized before, certain configurations, such as the 3f(T)-b-4f-234.7° site, had already been labeled as B5-sites [94] and shown to play a crucial role in ammonia synthesis [91] and decomposition [85] on Ru step surfaces. Four additional variants have been identified and labeled as 4f-b-3f(O)-200.7°, 3f(O)-b-4f-125.3°, 3f(T)-b-4f-164.2°, and 3f(T)-b-4f-125.3°, where “O” and “T” denote local octahedral and tetrahedral voids in ABAB (HCP) and ABCA (FCC) lattices, respectively. The angle appended to each label represents the geometric relationship between the equilateral triangle and square motifs that define the local adsorption site. For instance, the 3f(T)-b-4f site consists of a tetrahedral motif and a square pyramidal structure, connected through shared triangular facets. This configuration forms a distinct angle of 234.7°, giving rise to the specific designation 3f(T)-b-4f-234.7°. Such a site is located above a tetrahedral hole and can be found on both HCP and FCC NPs.

The C7 site is a catalytically active seven-atom cluster located at step edges or kinked regions of Fe surfaces, such as Fe(111) or Fe(211), which commonly arise in nanostructured or roughened iron catalysts. Here, “C” may stand for “cluster” or “coordination”, and “7” refers to the number of surface atoms involved in forming the site—specifically, seven iron atoms, depending on context. As such, the site consists of three atoms from the upper terrace, three from the lower terrace, and a single atom at the step edge, forming a unique geometry that offers undercoordinated iron atoms. This arrangement creates an ideal environment for nitrogen activation, making the C7 site a key player in the rate-limiting N_2_ dissociation step during ammonia synthesis [95,96]. To capture the diversity of iron surface structures, Zhang et al. [97] have explored low-index facets with Miller indices satisfying h + k + l ≤ 5 and h ≤ 3. In addition to the widely examined (111) and (211) facets, the (221), (311), (310), and (210) surfaces expose catalytically relevant C7 and/or B5 sites. Notably, the (111), (221), (311), and (211) surfaces all host C7 sites, with Wulff construction indicating that (211) contributes the largest surface area among them. Illustrations of the B5 and C7adsorption sites on the (311), (111), and (210) surfaces of FCC Co and BCC Fe (see Figure 1e–g).

Figure 1d illustrates the various adsorption configurations of molecular N_2_ often examined on transition metal surfaces [98]. These configurations—such as atop end-on (vertical), bridged end-on, bridged side-on (horizontal), and tilted—reflect the diversity of local coordination environments. The specific mode of N_2_ binding depends on the electronic structure and topology of the adsorption site, both of which vary across catalyst materials and surface terminations. These differences influence the binding strength and activation of N_2_, playing a critical role in catalytic performance.

## 3. The Activation Energy, *E*_a_

The activation energy (*E*_a_), a key kinetic parameter in ammonia synthesis, is widely examined in both experimental and theoretical studies [24,26,52,53,99,100,101,102,103,104]. It governs the rate constant *k* through the Arrhenius equation *k* = A *e*^−*Ea/RT*^, where A is the pre-exponential factor, which represents the frequency of molecular collisions, while the other symbols have their usual meaning. *E*_a_ is often discussed in relation to rate-determining or rate-limiting steps and corresponds to the energy required to transition from reactants to the high-energy transition state (TS). Overcoming this energy barrier is crucial for breaking the strong N≡N triple bond in molecular nitrogen and activating nitrogen atoms for subsequent reactions. It also applies to overcoming barriers in forming intermediates via hydrogenation and facilitating desorption, ultimately leading to ammonia production [105].

Despite nearly a century of advances in experiments, theory, and catalyst development, a complete mechanistic understanding of activation profiles and their relationship to the RDS in ammonia synthesis remains elusive—posing both a challenge and an opportunity for the rational design of next-generation catalysts. [106].

In heterogeneous catalysis, activation energy is influenced by the catalyst’s properties, including its electronic structure, surface morphology, and interactions with reactants. These factors facilitate adsorption, dissociation, and subsequent reaction of molecular species. Lower activation energy leads to a faster reaction rate, enhancing NH_3_ production and overall catalyst efficiency. For instance, the Ru SAs/S-1 catalyst [107] exhibits an apparent activation energy of 0.57 eV (55 kJ mol^−1^) for the reaction N_2_ + H_2_ → N-NH_2_ [77], as determined from an Arrhenius plot. Similarly, the *E*_a_ for N_2_ → 2N on Ca_3_CrN_3_H is 0.777 eV (75 kJ mol^−1^) [108], comparable to other hydride-based catalysts such as BaTiO_3−*x*_H*_x_* (80 kJ mol^−1^) [103], and BaCeO_3−*x*_N*_y_*H*_z_* (72 kJ mol^−1^) [109], and significantly lower than conventional Ru-based catalysts (85–121 kJ mol^−1^) [109]. *E*_a_ values for various transition-metal-based catalysts are given in Table 1.

Using the most stable N_2_ adsorption geometry as the initial state of an NEB simulation, Zhang et al. [97] theoretically examined N_2_ dissociation on eight Fe surfaces. Facets exposing C7 sites—(111), (221), (311), and (211)—that demonstrated consistently high activity, with activation energies narrowly ranging from −0.35 to −0.45 eV, in the order: (111) > (221) = (311) > (211). By contrast, (210) and (310), which lack well-defined C7 sites, showed lower activity (activation energies between −0.16 and −0.14 eV), while (100) and (110) exhibited the highest barriers (0.10 and 0.26 eV). These findings clearly highlight the critical role of active site geometry in governing N_2_ activation and, by extension, ammonia synthesis efficiency.

**Table 1 ijms-26-04670-t001:** Reported catalysts, rate-determining steps, reaction mechanism type, and apparent activation energies (wherever available), including the reaction mechanism, wherever feasible.

Catalyst	Rate-Determining Step	Mechanism	Activation Energy/eV
Ru	N_2_ → 2N	Dissociative	0.881–1.254 [108,110]
Ru/MgO; Ru/Al_2_O_3_	N_2_ → 2N	Dissociative	1.638 [62]
Fe (commercial)	N_2_ → 2N	Dissociative	0.725 [111,112]
Ru/Sm_2_O_3_	N_2_ → 2N	Dissociative	1.30 [113]
Co/Ba/La_2_O_3_	N_2_ → 2N		0.474–0.758 [114]
Co_3_Mo_3_N	N + H_x_ → NH_x_ (x = 1–3)	ER/MvK [115,116]	0.58 [27]
Ru/CeO_2_	N_2_ + H → N + NH	Associative	0.85 [117]
Ca_3_CrN_3_H	N_2_ + H → N=NH	Associative alternating	0.777 [108]
BaTiO_3−*x*_H*_x_*			0.829 [108]
BaCeO_3−*x*_N*_y_*H*_z_*			0.746 [108]
Cs−Ru/MgO			1.244 [117]
Ru/C12A7:*e*^−^	N + H_x_ → NH_x_ (x = 1–3)		0.508 [118]
Ru/[Ca_24_Al_28_O_64_]^4+^(O_2_^−^)_2_	N + H_x_ → NH_x_ (x = 1–3)		1.078 [75]
Ru/[Ca_24_Al_28_O_64_]^4+^(O_2_^−^)_2−x_ (*e*^−^)_2x_	N_2_ → 2N		0.518–0.622 [75]
Ru−Cs/[Ca_24_Al_28_O_64_]^4+^(O_2_^−^)_2_	N + H_x_ → NH_x_ (x = 1–3)		1.171 [75]
Ba-Co/C			0.954 [119]
Co@BaO/MgO; Co/Ba/MgO; Co/MgO	N_2_ → 2N		0.538; 0.803; 0.833 [120]
Ru/CaFH	N_2_ → 2N		0.207 [101]
Ru/Ba–Ca(NH_2_)_2_	N + H_x_ → NH_x_		0.425 [121]
LaCoSi	N + H_x_ → NH_x_ (x = 1–3)		0.435 [106]
Ru/Ca_2_N:*e*^−^			0.622 [24]
Ru–Cs/MgO			1.244 [24]
Ru/CaNH			1.14 [24]
Ru/CaH_2_			1.244 [24]
BaH_2_-BaO/Fe/CaH_2_			0.415 [122]
Ru/Ba−Ca(NH_2_)_2_			0.456 [121]
Ru/Ba–Ca(NH_2_)_2_			0.611 [102]
Ru/BaO–BaH_2_	N + H_x_ → NH_x_ (x = 1–3)	Anticipated from reaction order	0.425 [121]
Ru/La_0.5_Ce_0.5_O_1.75__650red	N_2_ → 2N		0.663 [123]
Ru/Ba–LaCeO_x__Ru_3_(CO)_12_	N_2_ → 2N		0.705 [124]
Ru/Ba–LaCeO_x__ALD	N_2_ → 2N		0.622 [124]
Ru/Ba–LaCeO_x__RuCl_3_	N_2_ → 2N		0.736 [124]
Ru-Ba/MgO			0.435 [125]
Ni/LaN; Ni/CeN NPs	N + H → NH		0.596 [27]; 0.556 [27]
Ru/CaH_1.72_O_0.14_			0.715 [126]
Ru/CaH_1.5_0O_0.25_			0.611 [126]
Ru/CaH_1.12_O_0.44_			0.829 [126]
CeNi_2_	N + H_x_ → NH_x_ (x = 1–3)	Anticipated from reaction order	0.573 [127]
CeNi_5_	N_2_ → 2N	Anticipated from reaction order	0.824 [127]
CeN NPs			0.586 [27]
Ru/BaAl_2_O_4−*x*_H*_y_*	N_2_ → 2N		0.681 [128]
Co/C			1.544 [111]
Ba_0.8_Co_1.0_/C			1.068 [111]
Co-Mo(5:5)/CeO_2_			0.591–0.632 [129]
10%Cs–FePc; FePc; CoPc	N_2_ → 2N	Dissociative	0.435; 0.434; 0.631 [112]
Co/BaAl_2_O_4−x_H_y_	N_2_ → 2N	Dissociative	0.514 [128]
Co/BaAl_2_O_4_	N_2_ → 2N	Dissociative	1.042 [128]
Fe/Ce_1−z_Sm_z_O_2x_N_y_	N_2_ → 2N	Dissociative	0.466 [130]
Fe/CeO^2−x^N_y_	N_2_ → 2N	Dissociative	0.518–0.684 [130]
Cr–LiH	N + H_x_ → NH_x_ (x = 1–3)		0.659 [131]
Mn–LiH	N + H_x_ → NH_x_ (x = 1–3)		0.524 [131]
Fe–LiH	N + H_x_ → NH_x_ (x = 1–3)		0.482 [131]
Co–LiH	N + H_x_ → NH_x_ (x = 1–3)		0.540 [131]

## 4. The Activation Energy Profile

In Figure 2a, the energy profile illustrates a typical case of N_2_ decomposition (N_2_ → 2N [132]) via the Langmuir–Hinshelwood (LH) mechanism [133], where the reaction enthalpy (Δ*H*) is exothermic. Here, Δ*H* is defined as Δ*H* = *E*(product) − *E*(reactant), with *E* referring to the total electronic energy of each individual state. On the surface of Fe, however, this reaction is endothermic, despite the overall ammonia synthesis reaction (N_2_ + 3H_2_ → 2NH_3_) being exothermic [134].

In Figure 2b an energy profile is shown where a promoter on the catalyst obstructs reactants, hindering their ability to find an immediate optimal pathway to the product state. The transition state (TS), representing the highest energy point, corresponds to the activation energy (*E*_a_).

Shown in Figure 2c is the profile that arises when a reactant, such as N_2_, undergoes an orientational transformation (e.g., from end-on to side-on) on the catalyst surface before forming the product state (2N). Here, *E*_a_’ represents the activation energy, and TS1 appears if the reactants pass through a quasi-stable intermediate state. This intermediate is likely to form when an active site near the reaction pathway influences the product before it reaches its most stable configuration at the most favorable active site on the catalyst surface.

For catalysts such as hexagonal Co or Ru, N_2_ is exothermically adsorbed in an end-on configuration atop a Co [135] or Ru [136,137] atom. By contrast, the sideways (or side-on) configuration at an hcp site is a well-activated but metastable state [65] characterized by weaker adsorption. For example, at the PW91 level of theory, the reported adsorption energies of N_2_ on Ru(0001) are −0.74 eV and −0.24 eV for top and hcp sites, respectively [138]. Zhang et al. [135] reported PBE-level adsorption energies for different sites, ranging from −0.46 to −0.67 eV for hcp Co and −0.42 to −0.61 eV for fcc Co. This behavior contrasts with the binding affinity of N_2_ at the vacant site on a Co_3_Mo_3_N(111) catalyst, where adsorption is endothermic [139,140] (*E*_ad_ = 0.415 eV [115], a trend similar to that observed for other metal nitrides, viz. Mn_6_N_5+x_ and η-Mn_3_N_2_ [99].

Assuming the side-on configuration corresponds to the reactant and the 2N configuration represents the product, the latter part of the reaction profile in Figure 2c would result, with *E*_a_ denoting the activation barrier. Most studies classify this metastable initial state as the reactant [141], and the activation energy between it and the product state is commonly reported [136]. As a result, the reaction energy profile is often depicted with a single activation barrier, rather than showing two distinct transition states. This explains why side-on N_2_ adsorption is frequently referenced in discussions of the LH mechanism [115]. Co_3_Mo_3_N serves as a representative system where this behavior is clearly observed [115].

N_2_ adsorbs in an end-on configuration before rotating into a side-on molecularly adsorbed state on CuNi(111) [142], following appropriate dissociatively adsorption states. Mortensen et al. [141] conducted a detailed DFT study on N_2_ adsorption and dissociation on Fe(111), identifying four distinct molecular adsorption states. Three were end-on configurations (perpendicular or tilted) at different adsorption sites, while the fourth was a side-on orientation, where both nitrogen atoms interacted with the surface. The side-on state was found to be more stable than the end-on state, with dissociation proceeding through it as a precursor. However, this behavior differed for N_2_ on a CuNi(111) catalyst, where end-on adsorption generally exhibited lower adsorption energy than the side-on mode [142].

The scenarios depicted in Figure 2d likely represent the hydrogenation process, where the reaction progresses through successive intermediate states. This interpretation is reasonable, as the formation of NH*_x_* (*x* = 2, 3) intermediates may be kinetically hindered on the catalyst surface (e.g., Ru) at low temperatures [143], while the NH intermediate remains relatively stable [144]. In this context, the product state is thermodynamically less favorable (Δ*H* > 0) than the reactant state, as observed for the Fe(111) catalyst in the reaction NH_2_ + H → NH_3_ [145].

The reaction energy profile in Figure 2f suggests that a dissociatively adsorbed H atom, initially stabilized at a thermodynamically favorable site (e.g., an hcp site), migrates to a less favorable site (e.g., an fcc site, a bridge, or an on-top site [146]) along the hydrogenation pathway before recombining with the N adatom to form NH*_x_* species. Some of the scenarios illustrated in Figure 2 were demonstrated in ref. [142] for N_2_ activation on a CuNi catalyst.

Abghouli et al. [147] reported that the adsorption of N_2_ molecules and N atoms on clean transition metal nitride surfaces is typically endothermic, indicating high energy demands in the initial stages. Moreover, direct N_2_ dissociation is hindered by substantial activation barriers exceeding 2 eV, posing a significant challenge for N_2_ activation. By contrast, hydrogen-assisted N_2_ activation via the associative Mars–van Krevelen (MVK) mechanism [148] (vide infra) is considerably more efficient, offering a viable alternative for facilitating N_2_ dissociation.

To determine the activation energy profiles and locate the transition states for N≡N bond cleavage, NH_x_ hydrogenation, and NH_3_ desorption, the Nudged Elastic Band (NEB) method [149] is commonly used [115,150], with vibrational analysis confirming the identification of a saddle point. NEB constructs a series of intermediate “images” between the initial and final states of a reaction pathway, connected by virtual springs, and optimizes them to trace the minimum energy path (MEP). The method ensures that the images relax properly on the potential energy surface by applying forces perpendicular to the path, while maintaining appropriate spacing along the reaction coordinate.

To refine the transition state estimate, the Climbing Image NEB (CI-NEB [151]) method is often employed, as demonstrated in studies of catalysts such as Fe_3_Mo_3_N [143], FeN_4_ [152], RuN_4_ [152], Co/MoC [153], and Ru-based TM@Ru(0001) (TM = Sc–Zn, Y–Cd) single-atom alloys [154], among others [155]. In CI-NEB, the image with the highest energy is driven uphill along the reaction coordinate and relaxed in all perpendicular directions, allowing it to converge to the saddle point, corresponding to the true transition state, without requiring prior knowledge of its geometry. The activation profiles shown in Figure 2a–f may result from applying NEB and/or CI-NEB to the reactant and product states associated with N_2_ dissociation and NHₓ formation.

The number of images used between reactant and product states typically ranges from three to ten [115,153,156]. For example, Zeinalipour-Yazdi et al. [115] first estimated the activation barriers associated with N_2_ dissociation and NH*_x_* formation by employing both dissociative and associative mechanisms, using 10 images to capture the features of the potential energy surface. Once key intermediates were identified and fully optimized, a subsequent NEB calculation with three images was performed to locate the transition state more efficiently for the Co_3_Mo_3_N catalyst [115].

## 5. Elementary Reaction Mechanisms and Reactant–Product Pairs for the Nitrogen Reduction Reaction (NRR)

Over time, various elementary reaction mechanisms for NRR have been proposed to elucidate the complex processes of adsorption, surface reactions, and desorption in catalysis [23]. These mechanisms include the conventional Haber–Bosch process [140,157,158], along with plasma-catalytic [159,160], photocatalytic [161], biocatalytic [162], and electrocatalytic approaches [163]. While these mechanisms can be dissociative, associative, or a combination of both [164], the last three primarily follow associative pathways for N_2_ fixation, typically operating at ambient pressure (1 atm) and low temperatures [165,166] with distinct RDS and minimum-energy pathways [156].

The H_2_ or N_2_ molecule from its gaseous state can adsorb on the catalyst at vacant surface sites [167] in different orientations—end-on, side-on, on-top bridge, on-top hollow, hollow sideways, or tilted [135,141,146,168]. After H_2_ dissociates on the catalyst, the chemisorption of atomic hydrogen onto the surface could occur with Δ*H* < 0, whereas absorption into the catalyst’s bulk is a process with Δ*H* > 0 [169]. Hydrogen species (either H_2_ or atomic H) can interact with adsorbed N_2_ or N species through direct contact from the gas phase or via surface activation.

A variety of N≡N bond dissociation steps have been suggested to explain nitrogen reduction reaction associated with the RDS [145,170,171], shown in Figure 3a–h.

If the initial stage of the reaction involves the dissociation of the N≡N triple bond (N_2_ → 2N; Figure 3a) under harsh conditions, a dissociative LH mechanism (vide infra) is typically considered operative [172,173]. This mechanism, first proposed by Langmuir in 1916 [174,175] and further developed by Hinshelwood in the late 1920s [176,177,178,179], depends on the catalyst’s nature, as the energy barrier for N_2_ dissociation can vary significantly. For instance, reported barriers range from 0.3 to 1.5 eV for different Miller index surfaces of Ru [180,181], 0.68 eV for Fe(111), 0.69 eV for K/Fe(111) [72,95], and between 0.55–1.37 eV for hcp Co and 0.64–1.39 eV for fcc Co [72,95].

If intermediate steps in N≡N dissociation involve partial hydrogenation, pathways like those in Figure 1b–f may arise. Back and Jung [171] demonstrated that reactant-product pairs such as N_2_→ 2N (Figure 3a), N–NH → N + NH (Figure 3b), and HN–NH → 2NH (Figure 3c) exhibit significant activation barriers, often exceeding 1 eV, rendering these steps kinetically hindered. While the latter two steps belong to the associative N_2_ fixation mechanism [165,166], Liu et al. [156] showed that the former step can occur after the first hydrogenation of N_2_ on Fe_3_/θ-Al_3_O_3_(010), leading to the dissociation of adsorbed N≡N into N and NH. This hydrogen-assisted N_2_ dissociation route is an associative process, also observed in Ru NPs [113,182] and Ru/CeO_2_ catalysts [117].

The structures of reactants, transition states, and products for N–N bond cleavage—illustrated in Figure 3a,b,e—were analyzed on the Fe_2_P(001) surface [183], revealing N_2_ dissociation as the rate-limiting step. Various reaction mechanisms, including direct dissociative adsorption and stepwise associative pathways, were considered to capture the diversity of possible activation modes.

Back and Jung [171] further demonstrated that the dissociation steps N–NH_2_ → N + NH_2_ (Figure 3d), NH–NH_2_ → NH + NH_2_ (Figure 3e), and NH_2_–NH_2_ → NH_2_ + NH_2_ (Figure 3f) have significantly lower activation barriers (<0.7 eV). These reduced barriers enhance the feasibility of these dissociation steps, making them more likely to occur under typical reaction conditions. This finding, along with results from other studies [115,150,184], may also extend to the reactant–product pairs in the reactions N–NH_3_ → N + NH_3_ (Figure 3g) and NH–NH_3_ → NH + NH_3_ (Figure 3h). In these cases, the favorable energetics of N–N dissociations [140] drive the overall reaction mechanism, promoting the reduction process at lower activation energies compared to the more challenging N≡N dissociation.

### 5.1. The Volcano Plot and Bronsted–Evans–Polanyi Relationship

The volcano plot is a visual representation that captures the relationship between catalytic activity—such as turnover frequency—and the binding energy of a specific reaction intermediate. In the context of ammonia synthesis, this relationship typically balances nitrogen adsorption—neither too weak (hindering activation) nor too strong (impeding desorption)—and thus centers around the chemisorption energy of nitrogen species (N or N_2_). A clear volcano-type trend, as illustrated in Figure 4a, has been established between nitrogen binding strength and catalytic activity [20], a relationship frequently referenced in catalyst screening studies [26]. This trend aligns with Sabatier’s principle: optimal catalysts exhibit intermediate nitrogen adsorption, strong enough to activate N_2_, but weak enough to permit subsequent hydrogenation and desorption. However, while useful for interpreting activity trends, this principle is often insufficient for guiding the rational design of optimal catalysts, as it overlooks the complexities of multistep reaction mechanisms and competing surface phenomena [185].

Metals on the left side of the volcano curve strongly adsorb nitrogen, leading to sluggish N–H bond formation and reduced ammonia synthesis rates. Conversely, metals on the right adsorb nitrogen too weakly, limiting N_2_ activation and constraining overall efficiency. Near the volcano peak, metals like Fe, and Ru exhibit intermediate nitrogen binding, with CoMo emerging as the most effective catalyst [186]. This optimal binding enhances nitrogen activation, facilitating efficient ammonia production. The superior performance of these metals is attributed to their favourable electronic properties, particularly the positioning of their d-band centres, which enable optimal interactions with nitrogen intermediates.

Nickel and cobalt generally exhibit lower catalytic activity compared to palladium-based systems. However, early transition metals such as Sc, V, Y, Ti, Zr, Nb, and Re demonstrate a stronger adsorption affinity for nitrogen relative to hydrogen, suggesting their potential suitability for nitrogen reduction reaction [171,187]. This preferential nitrogen binding behavior is probably advantageous in suppressing the competing hydrogen evolution reaction (HER), thereby enhancing selectivity toward ammonia production. Similar adsorption trends observed in thermocatalytic systems suggest that these metals could be promising candidates for both electrochemical and thermochemical ammonia synthesis. 

On the other hand, the Brønsted–Evans–Polanyi (BEP) relationship [188] provides a widely used framework in heterogeneous catalysis, linking activation energies to reaction enthalpies through linear scaling. In many transition metal systems for ammonia synthesis, this implies that a single descriptor—often the nitrogen adsorption energy—can approximate the energetics of key steps, such as N≡N dissociation and N–H bond formation. Specifically, stronger adsorption of the N atom implies a lower N_2_ dissociation barrier but higher NH*_x_* desorption energies, which is typically seen on metal surfaces like Re, Mo, and Fe. Conversely, weaker adsorption of the N atom results in a higher N_2_ dissociation barrier and lower NH*_x_* desorption energies [156]. Thus, a good metal catalyst for ammonia synthesis must exhibit a moderate atomic N adsorption energy, located around the peak of the volcano plot. However, when N_2_ hydrogenation becomes the dominant process, the N–N bond is significantly weakened, and the dissociation barrier no longer obeys the BEP relationship [189]. This alteration suggests that the BEP framework is more applicable in the early stages of the reaction and may not fully capture its complexity in the later stages, where different factors influence the reaction. Furthermore, the BEP relation also imposes a trade-off: optimizing one step (e.g., N_2_ activation) may adversely affect another (e.g., NH_3_ desorption), as seen with metals like Ru and Ni. To overcome this constraint, recent efforts have focused on breaking such scaling relations through catalyst architecture innovations, including bifunctional sites, support effects, and single-atom alloys [190]. For example, Ru single atoms on Cu (Ru@Cu) have demonstrated the decoupling of N_2_ activation and NH_3_ release, offering a promising route to circumvent BEP limitations and achieve enhanced catalytic performance.

A reaction energy diagram comparing two surface-catalyzed pathways for the transformation of a gas-phase reactant (A) to a gas-phase product (B) via adsorbed intermediates ([A], TS1, [B]) is presented in Figure 4b. In Pathway 1, the energy ordering is A(gas) < [A] < [B], indicating that adsorption stabilizes the molecule progressively. Despite this thermodynamic favorability, the reaction proceeds through a high-energy transition state (TS1), resulting in a large activation energy barrier (*E*_a_). In Pathway 2, both [A] and [B] are further stabilized (i.e., at lower energies than in Pathway 1), and the corresponding transition state (TS1′) lies at a lower energy, leading to a smaller activation barrier (*E*_a′_). This trend agrees with the Brønsted–Evans–Polanyi (BEP) relationship, which correlates the stability of intermediates with reduced kinetic barriers. Thus, Pathway 2 is both thermodynamically and kinetically more favorable.

While both the volcano plot and the BEP relationship [188] are central in catalysis, they offer different perspectives on catalyst performance. The volcano plot emphasizes the balance between nitrogen adsorption strength and catalytic activity, visualizing how intermediate binding energies correlate with turnover frequency. On the other hand, the BEP relationship links activation energies to reaction enthalpies through linear scaling, where a single descriptor—typically nitrogen adsorption energy—can approximate the energetics of key catalytic steps like N≡N dissociation and N–H bond formation. Both frameworks are useful, yet each has its limitations. The volcano plot is often too simplistic for optimizing complex multistep processes, while the BEP relationship, though insightful, can result in trade-offs [191] between different multistep reaction steps, such as the competing needs for efficient N_2_ activation and NH_3_ desorption [23].

Breaking the inherent scaling limitations in transition metal (TM)-based catalysts has been a major challenge, particularly when catalysts exhibit a scaling relationship between the binding energies of reaction intermediates. Recent advances in catalyst design have shown that introducing a second active site can help overcome this constraint [131,192,193,194]. Specifically, pairing low-valent metal centers can exhibit cooperative behavior, enhancing reaction pathways in either stepwise or concerted manners. This cooperative effect often leads to more efficient reactant activation, mitigating the typical scaling limitations seen in single-metal catalysts.

Liu et al. [195] have shown that the N_2_ dissociation barriers on Fe(111), Fe(211), Fe(110), and Fe(100) surfaces deviate from the conventional BEP trend. Specifically, Fe(111) exhibits both the weakest N_2_ adsorption and the lowest dissociation energy barrier. This unexpected behavior is linked to electron transfer from the iron surface to the π* antibonding orbital of adsorbed N_2_. Increased charge transfer into this orbital weakens the N≡N bond, thereby reducing the energy required to dissociate the N atoms. By contrast, the subsequent hydrogenation of N atoms and the desorption of NH_3_ on these surfaces still conform to the BEP principle. Consequently, Fe(111) emerges as the most catalytically active surface for ammonia synthesis—a trend that also extends to similar surfaces of nickel and molybdenum.

A promising example of this approach is the use of bimetallic single-cluster catalysts (SCCs), denoted as M_1_M′_n_ (e.g., Pt_1_Co_n_), which have gained attention for their ability to activate reactants more effectively. For instance, Ma et al. [192] studied NRR over Rh_1_Co_3_ clusters supported on CoO(011), finding that the reaction mechanism in this system follows an associative pathway, similar to the highly efficient enzymatic nitrogen fixation found in nature. This biomimetic mechanism illustrates the potential of SCCs to enhance thermal N_2_-to-NH_3_ conversion and highlights new strategies for overcoming the limitations of traditional single-metal catalysts.

Fe_3_ clusters supported on θ-Al_2_O_3_(010) activate N_2_ via an associative *NNH intermediate rather than the traditional dissociative pathway, reducing the energy barrier beyond BEP expectations [196]. Phosphorus-modified Fe_2_P(001) surfaces exhibit multiple N–N activation routes and altered NH*_x_* binding energies, defying the typical scaling behavior [183]. In Co/BaCeO_3_ catalysts modified with yttrium, the enhanced performance could not be explained solely by N_2_ adsorption energies, indicating additional electronic or structural factors are at play [197], which are not fully explained by traditional BEP scaling. Collectively, these studies demonstrate that while BEP offers valuable predictive insight, it often breaks down in the presence of defects, support effects, alternative mechanisms, or under non-equilibrium reaction conditions. This emphasizes the need for a more flexible framework in catalyst design.

### 5.2. The Dissociative Langmuir–Hinshelwood (LH) Mechanism

Langmuir envisioned the reaction as unfolding through a series of dynamic, striking collisions [175,198], conceptualized in the following stages [199]:(i)Reaction between adsorbed species and the surface: adsorbed species interact with the underlying surface through processes such as migration or diffusion [200]. These interactions modify surface structure and reactivity, influencing both reaction kinetics and catalytic performance.(ii)Collision between gas molecules and adsorbed species: gas-phase molecules (e.g., H_2_) collide with adsorbed species (e.g., N or N_2_), transferring energy to surface-bound atoms and triggering catalytic transformations.(iii)Interaction between adsorbed molecules/atoms: this occurs when adsorbed species on the surface interact with each other in adjacent spaces on the catalyst, forming intermediates that are crucial for the reaction.

Many studies [200,201,202] illustrate stage (iii), particularly in hydrogenation steps [62,203] such as N + H ⇌ NH, NH + H ⇌ NH_2_, and NH_2_ + H ⇌ NH_3_. The process culminates in NH_3_ desorption from the catalyst surface [204,205]. This associative sequence also defines the MvK mechanism [148]. By contrast, the LH mechanism follows a dissociative pathway [7,206,207] where reactants undergo dissociative chemisorption before interacting on neighboring sites, ultimately leading to NH_3_ desorption [62].

Studies by Honkala et al. [110] and Strongin et al. [208] highlight the role of Ru’s B5 sites [110] and Fe’s C7 sites [208] as key active centers for N_2_ adsorption and dissociation. Under ambient conditions, both N_2_ and H_2_ adsorb molecularly onto the catalyst surface [140,146,206,209,210], reaching equilibrium with the catalyst [199]. N_2_ adopts a side-on orientation to enhance triple bond activation, while H_2_ follows a similar pattern. Under harsh conditions, dissociative adsorption occurs, facilitated by electron density transfer from the catalyst substrate [211].

The Horiuti–Polanyi (HP) mechanism is a dissociative mechanism [212], primarily involving the homolytic (or heterolytic) dissociation of H_2_ on the catalyst surface [213], such as Ag(211) and Ni(111) [214,215], with the dissociation step being the rate-determining step [216]. In the case of homolytic cleavage, the H_2_ molecule splits evenly; each H adsorbs onto the catalyst before participating in hydrogenation. In the case of heterolytic cleavage, H_2_ splits unevenly, with one hydrogen atom adsorbing onto the metal surface, while the other adsorbs onto a heteroatom (such as nitrogen) on the catalyst [216].

The ammonia synthesis reaction, N_2_ + 3H_2_ ⇌ 2NH_2_, proceeds via a series of elementary steps, a-o, on the catalyst surface, characteristic of the LH mechanism and illustrated in Figure 5a–n.
(a)[] + N_2_(g) + 3H_2_(g)Reference state: clean catalyst, unreacted N_2_(g) and H_2_(g)(b)[] + N_2_(g) ⇌ [N_2_]Adsorption of molecular N_2_ from gas phase(c)[N_2_] ⇌ [N + N]Dissociative adsorption of N_2_(d)[N + N] + H_2_(g) ⇌ [N + N + H + H]Dissociative adsorption of first H_2_(g)(e)[N + N + H + H] ⇌ [N + NH + H]Formation of first -NH intermediate(f)[N + NH + H] ⇌ [N + NH_2_]Formation of first -NH_2_ intermediate(g)[N + NH_2_] + H_2_(g) ⇌ [N + NH_2_ + H + H]Dissociative adsorption of second H_2_(g)(h)[N + NH_2_ + H + H] ⇌ [N + NH_3_ + H]Formation of first NH_3_ intermediate(i)[N + NH_3_ + H] ⇌ [N + H] + NH_3_(g)Desorption of first NH_3_(j)[N + H] ⇌ [NH]Formation of second -NH intermediate(k)[NH] + H_2_(g) ⇌ [NH + H + H]Dissociative adsorption of third H_2_(g)(l)[NH + H + H] ⇌ [NH_2_ + H]Formation of second -NH_2_ intermediate(m)[NH_2_ + H] ⇌ [NH_3_]Adsorption of second NH_3_ intermediate(n)[NH_3_] ⇌ [] + NH_3_(g)Desorption of second NH_3_(o)[] + 2NH_3_(g) → []Desorption of both NH_3_ and leaving clean catalyst

The reference state (a) corresponds to the clean catalyst surface, [M], with active sites M, along with one mole of N_2_ and three moles of H_2_ in the gas phase, i.e., [M] + N_2_(g) + 3H_2_(g). In step (b), N_2_ adsorbs from the gas phase onto the surface. Step (c) involves the dissociative adsorption of N_2_, where the N≡N bond is cleaved to form two surface-bound nitrogen atoms. In step (d), the first H_2_ molecule undergoes dissociative adsorption, forming two surface-bound H atoms. Subsequent hydrogenation steps lead to the formation of a metal–imide intermediate (M–NH; step e), then a metal–aminyl (M–NH_2_; step f). Step (g) involves adsorption and dissociation of a second H_2_ molecule, enabling further hydrogenation to a metal–ammonia species (M–NH_3_; step h). In passing from step (h) to step (i), the first NH_3_ molecule desorbs into the gas phase. The second N atom undergoes an analogous sequence: formation of M–NH (step j), followed by dissociative adsorption of a third H_2_ molecule (step k), leading to M–NH_2_ (step l) and M–NH_3_ (step m). The second NH_3_ molecule desorbs in step (n), and step (o) represents the final state, where both ammonia molecules have desorbed, regenerating the clean catalyst surface [M]. Clearly, each N atom follows a similar sequence of hydrogenation steps, but the exact path may slightly vary depending on the specific intermediate and surface site involved.

Often, the symbol “*” is used to denote an active site on a catalyst surface (in contrast to our use of “[]” to represent the clean catalyst that has the active site M), with reactions written as *2 + N_2_(g) ⇌ *2N (rather than that of step b, for example) or *2 + H_2_(g) ⇌ *2H [217]. These expressions imply that the adsorbed molecule dissociates into two atoms, each binding to a separate active site—an assumption valid when the molecule adopts a side-on (horizontal) orientation, allowing simultaneous interaction with two adjacent sites. However, this representation is less accurate for end-on adsorption, where the molecule initially binds through one atom to a single (atop) site on the catalyst (see atop adsorption of N_2_ in Figure 1d). Nevertheless, after cleavage, the resulting atoms may still occupy two distinct active sites through dissociative adsorption.

Transition metal nitrides, such as Ta_3_N_5_(100) [140,150] and η-Mn_3_N_3_(100) [218], serve as catalysts for elucidating the LH mechanism in ammonia synthesis. The adsorption of N_2_ at nitrogen vacancies is moderately endothermic, with an energy of 0.199 eV. Strong ammonia adsorption (*E*_ad_ = 2.332–2.934 eV), comparable to its desorption energy, has been identified as the RDS, indicating that high temperatures are required for effective ammonia synthesis.

Catalysts such as Co_3_Mo_3_N(111) [115] and Mo-terminated δ-MoN(0001) [211] may facilitate ammonia synthesis at elevated temperatures via the LH mechanism. However, for Co_3_Mo_3_N, this pathway is less kinetically favorable due to higher energy barriers in the hydrogenation steps [150].

Ru catalysts supported by electrides, such as as [Ca_24_Al_28_O_64_]^4^⁺(*e*^−^)_4_ and Ca_2_N:*e*^−^, exemplify systems where LH-based rate equations aid in determining reaction kinetics [25].

The associative and dissociative concerted mechanism proposed by Ye et al. [99] for Co/CeN catalysts aligns with the LH model for N_2_ dissociation, while also highlighting the role of nitrogen vacancies in facilitating the associative process [116]. Similarly, Co or Fe atoms enhance N_2_ dissociation on the electron-rich surface of molybdenum carbide (Mo_2_C), forming N_ad_ atoms in accordance with the LH mechanism [57]. It has been suggested that the RDS shifts from nitrogen dissociation to the hydrogenation of surface-bound NH_x_ species. However, whether the rate-limiting step and the rate-determining step are identical remains unclear, as the former is the slowest step in the sequence, while the latter controls the overall reaction rate. Furthermore, the specific hydrogenation intermediate corresponding to the RDS after this shift was not explicitly identified.

Mo-terminated γ-Mo_2_N(111) and δ-MoN(0001) catalysts are nearly identical, both exhibiting exothermic adsorption of N_2_, with adsorption energies ranging from −1.07 to −2.54 eV for δ-MoN(0001) at the PBE functional level of theory [211]. Notably, the side-on adsorption at the bridge site is favored over other adsorption sites, such as on top or tilted molecule orientation. The authors proposed that the rhombic configuration of the nearest-neighbor Mo sites exists on Mo-terminated γ-Mo_2_N(111), while near-identical surface site configurations are observed on Mo-terminated δ-MoN(0001). Moreover, the well-established method for controllable growth of single-crystalline, hexagonal MoN thin films provides a foundation for developing effective strategies to sustain Haber–Bosch catalysis by the LH mechanism with these materials.

N_2_ dissociation on pristine δ-MoN(0001) is associated with an *E*_a_ of 0.52 eV, but the reaction is rate-limited by the subsequent hydrogenation step, with an *E*_a_ of 2.00 eV for NH + H → NH_2_. By contrast, the activation energies for the reactions N + H → NH (*E*_a_ = 1.42 eV) and NH_2_ + H → NH_3_ (*E*_a_ = 2.00 eV) are lower. These values are different to those estimated for the corresponding reactions on γ-Mo_2_N(111), where *E*_a_ values are 0.58, 1.18, 1.47, and 1.28 eV, respectively [186,211]. However, in this case, the RDS is associated with the formation of an NH_2_ intermediate and the desorption of NH_3_. This contrasts with N-terminated γ-Mo_2_N, where the RDS is N_2_ → 2N, and the hydrogenation steps have activation barriers in the range 1.2–1.6 eV, with NH_3_ desorption having an activation energy of 0.87 eV. It was further shown that a significantly larger activation barrier can be observed for the hydrogenation step on the γ-phase model (*E*_a_ = 2.56 eV) for NH_2_ + H → NH_2_, particularly when the system’s initial state is arbitrary.

Sato et al. [120] proposed that N_2_ adsorption on Co nanoparticles supported on MgO weakens the N≡N triple bond to the strength of a double bond. This weakening was attributed to electron donation from Ba^2+^ in BaO, mediated through neighboring Co atoms, which facilitates N_2_ bond cleavage. They proposed the dissociation step (N_2_ → 2N) as the RDS in ammonia synthesis. However, this conclusion was based on assumptions rather than a thorough examination of reaction mechanisms and adsorption characteristics, analogously as conducted in studies on lanthanoid-oxide-supported Ru [219] and Fe/Ba/MgO [47] catalysts. More recently, Miyazaki et al. [52] challenged this perspective, demonstrating that N_2_ dissociation—as in Ru-based catalysts [62]—is not the RDS in most catalytic systems, including Fe/BaTiO_3−*x*_N*_y_*. Instead, the breakdown of scaling relations suggests that catalytic activity is governed by the adsorption of intermediates and transition-state energies on the transition metal surface.

### 5.3. The Associative Langmuir–Rideal and Eley–Rideal Mechanisms

The Langmuir–Rideal (LR) mechanism [175], an adsorption-abstraction process in heterogeneous atom recombination [220], involves a gas-phase reactant (atom or molecule) directly colliding with an adsorbed species on the catalyst surface, as per condition (ii) (Section 5.2). Under ambient conditions (1 atm), such collisions are rare. However, an atomic gas-phase reactant may form via the hot-atom mechanism if its precursor molecule undergoes multiple rebounds on the surface [221,222]. The key distinction between the LR and LH mechanisms lies in the thermal equilibrium: the gaseous reactant in the LR mechanism is not in thermal equilibrium with an adsorbed species on the catalyst surface [199].

The Eley–Rideal (ER) mechanism [223,224], often referenced in plasma catalysis [159,160], operates in a low-entropy regime for the gas-phase reactant. Unlike the LR mechanism, the ER mechanism entails the gas-phase reactant being weakly adsorbed onto the catalyst surface through van der Waals interactions [225]—essentially a form of physisorption [223]. This weak adsorption mitigates the disorder of the gas-phase species, as compared to the LR mechanism, reducing the entropic factor [188]. The rest of the ER mechanism is driven by collisional interactions between the physisorbed species and the chemisorbed surface species, facilitating the hydrogenation steps. A detailed comparison of the LR and ER mechanisms can be found elsewhere [198,220].

The N_2_ molecule undergoes chemisorption on the catalyst surface, while the reactant H_2_ may be physisorbed, particularly when the surface exhibits limited reactivity (inertness) toward its adsorption. In such cases, the surface is ineffective in cleaving the H–H bond [213], or H_2_ remains in the gas phase when the surface fails to provide the necessary site-specific environment for physisorption. The former occurrence is not uncommon on transition metal surfaces such as Cu, Ag, and Au [215].

The collision between the interacting species dictates whether the LR or ER mechanism governs the formation of NH_3_ [200,216]. The reaction follows an associative mechanism, where recombination between reactants occurs. Some studies suggest that both the LH and ER mechanisms can contribute collectively to the overall reaction rate [226]. Notably, Elis et al. [227] demonstrated that the ER mechanism dominates at very high temperatures (around 1250 K) for specific catalysts, in contrast to the LH mechanism.

The elementary steps (a–l) representing the Eley–Rideal mechanism for the reaction N_2_(g) + 3H_2_(g) ⇌ 2NH_3_(g) on the catalyst surface are outlined below.
a.[] + N_2_(g) + 3H_2_(g)Reference state: clean catalyst; unreacted N_2_(g) and H_2_(g)b.[] + N_2_(g) ⇌ [N_2_]Chemically adsorbed N_2_ moleculec.[N_2_] + H_2_(g) ⇌ [N-N + H_2_]Physically adsorbed first H_2_ moleculed.[N-N + H_2_] ⇌ [N-NH_2_]Formation of N-NH_2_ intermediatee.[N-NH_2_] + H_2_(g) ⇌ [N-NH_2_ + H_2_]Physically adsorbed second H_2_ moleculef.[N-NH_2_ + H_2_] ⇌ [NH-NH_3_]Formation of NH-NH_3_g.[NH-NH_3_] ⇌ [NH] + NH_3_(g)Desorption of first NH_3_(g)h.[NH] + H_2_(g) ⇌ [NH + H_2_]Physisorption of third H_2_ moleculei.[NH + H_2_] ⇌ [NH∙∙∙H_2_]Formation of NH∙∙∙H_2_ weakly bonded intermediatej.[NH∙∙∙H_2_] ⇌ [NH_3_]Formation of second NH_3_k.[NH_3_] ⇌ [] + NH_3_(g)Desorption of second NH_3_l.[] + 2NH_3_(g) ⇌ []Desorption of two NH_3_; clean catalyst regenerated

The elementary steps (a’-l’) of the LR mechanism for the reaction N_2_(g) + 3H_2_ (g) ⇌ 2NH_3_(g) on the catalyst surface [] are as follows:(a’)[] + N_2_(g) + 3H_2_(g)(b’)[] + N_2_(g) ⇌ [N_2_](c’)[N_2_] + H_2_(g) ⇌ [N-N∙∙∙H_2_](d’)[N-N∙∙∙H_2_] ⇌ [N-NH_2_](e’)[N-NH_2_] + H_2_(g) ⇌ [N-NH_2_∙∙∙H_2_](f’)[N-NH_2_∙∙∙H_2_] ⇌ [NH-NH_3_]/[NH_2_-NH_2_]      (/ refers another possibility)(g’)[NH-NH_3_]/[NH_2_-NH_2_] + H_2_(g) ⇌ [NH-NH_3_∙∙∙H_2_]/[NH_2_-NH_2_∙∙∙H_2_](h’)[NH-NH_3_∙∙∙H_2_]/[NH_2_-NH_2_∙∙∙H_2_] ⇌ [NH_3_∙∙∙NH_3_](i’)[NH_3_∙∙∙NH_3_] ⇌ [NH_3_] + [NH_3_](j’)[NH_3_] + [NH_3_] ⇌ [NH_3_] + NH_3_(g)(k’)[NH_3_] + NH_3_(g) ⇌ [] + NH_3_(g)(l’)[] + 2NH_3_(g) → []

The key aspects of the minimum-energy pathway following the LR and ER mechanisms are illustrated in Figure 6. Both mechanisms can operate when N_2_ adopts an end-on (or indeed side-on) orientation [150]. While not explicitly shown for both, the end-on mode (left, a–k) is depicted for the ER mechanism, and the side-on (right, a′–k′) for the LR mechanism.

For the ER mechanism, the process begins with a clean catalyst [M] (a), which facilitates the chemisorption of N_2_ in step (b). The first H_2_ molecule is physisorbed onto the same catalyst and non-covalently interacts with the distal N atom of chemisorbed N_2_ in step (c), forming a hydrazine intermediate (N-NH_2_) in step (d). A second H_2_ molecule is adsorbed in a similar manner in step (e), and reacts to form the NH–NH_3_ species (see step f). The first NH_3_ molecule desorbs when passing from (f) to (g), leaving the NH intermediate in step (g). A third H_2_ molecule is physisorbed in step (h), forming a H_2_∙∙∙NH intermediate (i). The second NH_3_ molecule desorbs in step (k), and step (l) represents the final state in which both ammonia molecules have desorbed, fully regenerating the clean catalyst surface [M] as shown in (a).

For the LR mechanism, when the N_2_ molecule is adsorbed in its side-on orientation (b′), the first H_2_ molecule interacts directly with one of the N atoms via non-covalent interactions (c′). Alternatively, if N_2_ adopts an end-on adsorption geometry (as in step b), the distal nitrogen atom is the primary site of interaction with the incoming H_2_ molecule. A hydrazine (N=NH_2_) intermediate (d′) is formed regardless of the adsorption mode. A second H_2_ molecule adsorbs (e′), progressing to an intermediate (f′). Two possible interaction modes are shown in steps e′–g′, leading to NH_2_-NH_2_ or NH-NH_3_ intermediates (f′). The N-N bond may break at this stage, or if it remains intact, a third H_2_ molecule interacts with the intermediate, advancing from (f′) to (g′). This leads to stage (h′), where two NH_3_ species are non-bonded to each other. Stage (i′) follows, where the NH_3_ species are physically separated, either via chemisorption or physisorption. The first NH_3_ molecule desorbs from stage (i′) to (j′), and the second NH_3_ molecule desorbs from (j′) to (k′). Step (l′) represents the final stage, where both NH_3_ molecules have desorbed, regenerating [M]. While the pathway is hypothetical, the actual state of the intermediate species may vary depending on the nature of the catalyst surface.

Lan et al. [145] investigated the LH and ER mechanisms on an Fe(111) surface, identifying NH_2_ + H → NH_3_ as the rate-determining step. Their DFT calculations revealed that the top shallow site on Fe(111) is the most energetically favorable position for NH_2_. For the ER mechanism, they simulated the migration of a surface-activated H atom, placing it in a vacuum approximately 4.6 Å above the Fe surface. As the H atom approached the surface, it descended and reacted with the adsorbed NH_2_, thereby forming NH_3_.

While the authors attributed surface-adsorbed H migration to the LH mechanism and vacuum-phase H migration to the ER mechanism, the latter process, in our view, aligns more closely with the LR mechanism. However, the study does not explicitly confirm whether the H atom first adsorbs onto the Fe surface before reacting with NH_2_. Temporary interaction or weak adsorption before reaction is common in the ER mechanism. Despite this ambiguity, the study concluded that the ER and LH mechanisms share close similarities, with the key distinction being the vertical position of the H atom: in the ER mechanism (more accurately, the LR mechanism), the H atom descends toward NH_2_, whereas in the LH mechanism, it remains elevated as NH_3_ forms.

It was argued that the θ-Mn_6_N_5_-(111) catalyst facilitates an ER mechanism, but its kinetics are unfavorable due to the high barrier for surface nitrogen hydrogenation [150]. The nitrogen vacancies on the catalyst surface play a crucial role in driving the reaction, favoring an associative pathway [218]. Rouwenhorst et al. [228] proposed a similar ER mechanism involving comparable intermediate steps. However, in their model, association occurs through direct interaction between dissociatively adsorbed hydrogen atoms and N radicals from the plasma on MgO-supported Ru, Co, Pt, Pd, Cu, and Ag catalysts. In this process, the reaction N + H → NH on the catalyst surface was identified as the rate-limiting step. Since the N radicals originate from the plasma and are not physically adsorbed, this mechanism, in our view, aligns more closely with the LR mechanism rather than the ER mechanism.

N_2_ adsorbs more strongly than H_2_ on Mn_6_N_5+*x*_ (*x* = 1)-(111) catalysts [229]. When adsorbed in an end-on configuration, N_2_ exhibits minimal activation. Hydrogenation via the ER mechanism faces high energy barriers (>1.866 eV or 182 kJ mol^−1^), making ammonia synthesis impractical on this catalyst unless elevated temperatures are applied. A similar conclusion was drawn for the η-Mn_3_N_2_-(100) catalyst, where N_2_ adopts a side-on configuration in both the ER and LH mechanisms [215].

Zhang et al. [230] demonstrated that Ru clusters on MgO exhibit significantly stronger N_2_ adsorption (*E*_ad_ = −1.95 eV) compared to Ru/SiO_2_ (*E*_ad_ = −0.75 eV) and Ru/Al_2_O_3_ (*E*_ad_ = −1.33 eV). N_2_ dissociation on Ru/MgO has a low barrier (1.07 eV) and is exothermic by 0.33 eV. However, the high activation energy for hydrogenating surface-adsorbed N and NH*_x_* species hinders NH_3_ formation via the LH mechanism, particularly in the NH_2_ + H → NH_3_ step, leading to a lower production rate. Instead, NH_3_ production via the ER mechanism was suggested in a plasma environment, involving reactions such as H_2_(g) + NH or H(g) + NH_2_, with the hot-atom mechanism assumed to aid N_2_ dissociation. However, the process corroborates more accurately with the LR mechanism rather than the ER mechanism.

### 5.4. The Associative Mars–Van Krevelen (MvK) Mechanism

The Mars–van Krevelen mechanism [148] involves a redox process in which the catalyst surface undergoes oxidation and reduction cycles during ammonia synthesis. In this process, the first NH_3_ molecule is desorbed via hydrogenation of metal-coordinated N sites as in the backbone of the catalyst and a nitrogen vacancy is created through the reduction of mono-, bi- or tertiary- metal nitrides MN (e.g., TiN, CeN, Mo_2_N and Co_3_Mo_3_N) [104]. Introducing surface nitrogen vacancies, which are replenished by the direct capture of externally supplied gaseous N_2_, facilitates its subsequent activation. The associative distal-type pathway (vide infra) becomes the primary mechanism, where the distal N atom in N_2_ undergoes hydrogenation before the dissociation of the N≡N triple bond [42]. This leads to the formation and desorption of the second NH_3_ molecule, leaving behind the clean metal nitride catalyst [231], as observed for A–Mn–N (A = Li, K, Fe or Co) materials [232].

The elementary reaction pathway corresponding to the MvK mechanism, progresses through the following steps (a–n), where “•” refers to the vacancy site that represents the active absorption site on the clean catalyst [].
a.[•] + N_2_(g) + 3H_2_(g)Reference: clean catalyst with vacancy, unreacted N_2_ (g) and H_2_(g)b.[•] + N_2_(g) ⇌ [N_2_]Molecular adsorption of N_2_(g) at the vacancyc.[N_2_] + H_2_(g) ⇌ [N_2_ + H_2_]Molecular adsorption of H_2_d.[N_2_ + H∙∙∙H] ⇌ [N-NH + H]Dissociation of H_2_, formation of N−NH intermediatee.[N-NH + H] ⇌ [N-NH_2_]Formation of N-NH_2_ intermediatef.[N-NH_2_] + H_2_(g) ⇌ [N-NH_2_ + H_2_]Molecular adsorption of second H_2_g.[N-NH_2_ + H∙∙∙H] ⇌ [N-NH_3_ + H]Dissociation of H_2_ and formation of first NH_3_h.[N-NH_3_ + H] ⇌ [N H] + NH_3_(g)Desorption of first NH_3_(g)i.[N H] ⇌ [NH]Formation of second NH intermediatej.[NH] + H_2_(g) ⇌ [NH + H_2_]Molecular adsorption of third H_2_k.[NH + H∙∙∙H] ⇌ [NH_2_ + H]Dissociation of H_2_ and formation of NH_2_l.[NH_2_ + H] ⇌ [NH_3_]Formation of second NH_3_m.[NH_3_] ⇌ [•] + NH_3_ (g)Desorption of second NH_3_(g)n.[•] + 2NH_3_(g) ⇌ [•]Desorption of two NH_3_, leaving clean catalyst

A schematic view of the entire reaction pathway for the MvK mechanism is shown in Figure 7a–m, following a consecutive pathway for protonation [162]; the term “consecutive” refers to the step-by-step hydrogenation of the same nitrogen atom in N_2_. The reference state (a) consists of the clean catalyst surface with an oxygen vacancy, unreacted N_2_(g), and three H_2_(g) molecules. The reaction pathway proceeds via a surface with a vacancy or defect (b), where N_2_ initially adsorbs—typically in an end-on, tilted, or side-on configuration (the first two depicted in b)—at the active site [206,233]. This is followed by H_2_ adsorption (c), in line with the mechanistic description reported by Jesudass et al. [231].

The dissociation of H_2_ begins to occur at this stage, with one hydrogen atom transferring to the distal nitrogen atom in N_2_, leading to the formation of the N-NH intermediate (d). At this point, N_2_ may remain in its original end-on titled configuration (as on the left) or shift to a side-on orientation (as on the right). Continued hydrogenation results in the formation of the N-NH_2_ intermediate (e).

The addition of a second H_2_ molecule leads to the configurations depicted in step (f). The reaction pathway continues with the formation of a diazane intermediate in step (g), which facilitates the desorption of the first NH_3_ molecule as the system transitions from step (g) to (h). The subsequent steps (h–l) proceed in a manner analogous to the associative part of the LH mechanism [233,234] (see Figure 5i–m), involving the sequential formation of NH, NH_2_, and NH_3_ intermediates. This culminates in the desorption of the second NH_3_ molecule in step (m) of Figure 7, restoring the clean catalyst with the vacancy to complete the catalytic cycle (step n.).

It should be noted that the dissociation of the N≡N bond can occur at any point between steps (d) and (g), depending on the reactivity of the transition metal catalyst, although this is not explicitly depicted in the schematic. While this study primarily focuses on the use of molecular N_2_ and H_2_, other studies (e.g., (viz. ref. [147]) explore the stepwise hydrogenation of a single H atom to the distal nitrogen site in N_2_ on specific metal mononitride catalysts (such as rocksalt(100)) for N_2_ electroreduction and ammonia pair formation.

An ER-MvK hybrid mechanism has been proposed for ammonia synthesis on catalysts like molybdenum nitride (Co_3_Mo_3_N) [159], iron-molybdenum nitride (Fe_3_Mo_3_N) [143,150], and other transition metal nitride catalysts [196], combining elementary steps from both the ER and MvK mechanisms. In this model, N_2_ adsorbs onto the catalyst in a side-on [143] or end-on orientation [101,115,229], where the distal N site first interacts with gaseous H_2_ directly [235], forming an -N-NH_2_ intermediate on the Fe_3_Mo_3_N catalyst (similar to that shown in Figure 7d (right)). In another study [115], the same authors demonstrated that the first H_2_ molecule directly interacts with the distal N atom of end-on adsorbed N_2_, forming a trans-hydrazine intermediate on the Co_3_Mo_3_N(111) catalyst, and the subsequent steps follow the MvK pathway, as shown in Figure 7f–m. Even though it’s called the ER-MvK mechanism for Fe_3_Mo_3_N [143] and Co_3_Mo_3_N [115] catalysts, it results from a combination of the LR and MvK mechanism, and could be referred to as an LR-MvK mechanism instead.

The elementary pathway associated with the LR-MvK mechanism may proceed through steps (a–m, below). Most steps follow the typical ER-MvK mechanism, except for step c ([N_2_] + H_2_(g) ⇌ [N-NH_2_]) that involves direct interaction between adsorbed N_2_ and gaseous H_2_. The remaining two H_2_ molecules are initially physisorbed on the surface, rather than immediately reacting with the chemisorbed N_2_.
a.[•] + N_2_(g) + 3H_2_(g)Reference state: clean catalyst with a vacancy, unreacted N_2_(g) and H_2_(g)b.[•] + N_2_(g) ⇌ [N_2_]Molecular adsorption of N_2_(g) at the vacancyc.[N_2_] + H_2_(g) ⇌ [N-NH_2_]LR step: first H_2_ interacts directly with distal N in N_2_, forming N-NH_2_d.[N-NH_2_] + H_2_(g) ⇌ [N-NH_2_ + H_2_]Adsorption of second H_2_ moleculee.[N-NH_2_ + H_2_] ⇌ [N-NH_3_ + H]Formation of N-NH_3_ intermediatef.[N_2_H_3_ + H] ⇌ [N + NH_3_ + H]Formation of first NH_3_g.[N + NH_3_ + H] ⇌ [N H] + NH_3_ (g)Desorption of first NH_3_(g)h.[N + H] ⇌ [NH]Formation of first NH intermediatei.[NH] + H_2_(g) ⇌ [NH + H_2_]Adsorption of third H_2_ moleculej.[NH + H_2_] ⇌ [NH_2_ + H]Formation of NH_2_ intermediatek.[NH_2_ + H] ⇌ [NH_3_]Formation of second NH_3_l.[NH_3_] ⇌ [•] + NH_3_ (g)Desorption of second NH_3_(g)m.[•] + 2NH_3_(g) ⇌ [•]Desorption of two NH_3_ molecules, leaving behind the clean catalyst

Roy and Kumar [57] suggested that both the LH and MvK mechanisms can coexist on Mo_2_C, Fe/Mo_2_C, and Co/Mo_2_C thermo-catalysts, each contributing to ammonia synthesis. However, the Co/Mo_2_C catalyst exhibited the higher performance, attributed to its strong electron-donating ability, which enhances nitrogen dissociation For instance, at 520 °C and 1 bar, the specific activities for Mo_2_C, Fe/Mo_2_C, and Co/Mo_2_C were reported to be 8.58, 9.78, and 11.73 μmol h^−1^ m^−2^, respectively.

Kitano et al. [109] found that BaCeO_3−*x*_N*_y_*H*_y_* acts as an efficient thermo-catalyst operating through the MvK mechanism, with lattice N^3−^ and H^−^ ions mediating the process. However, ammonia synthesis typically occurs over conventional catalysts via the LH mechanism, highlighting the combined role of the MvK-LH model. Zhang et al.’s review [236] provides a detailed account of this mechanism on catalysts, such as Ru/LaN/ZrH_2_ [237].

Due to the high activation barrier for N_2_ dissociation and the endothermic adsorption of N_2_ on clean transition metal nitride surfaces, the MvK mechanism is a more favorable electrocatalytic NRR pathway [147,238]. Abghoui et al. [147,239] identified VN, CrN, ZrN, and NbN as the most promising catalysts, with the rocksalt (100) facet exhibiting the highest activity. These materials, due to their stable N vacancies and resistance to poisoning by −H or −O, demonstrated superior NRR activity compared to HER. Guan et al. [240] also highlighted the significance of the MvK mechanism in various vacancy-assisted catalysts.

Kobayashi et al. [103] demonstrated that at an elevated temperature of 400 °C, solid-state hydride-containing titanium compounds—specifically TiH_2_ and BaTiO_2.5_H_0.5_—give rise to a nitride-hydride surface strikingly similar to that observed in titanium clusters. Under the influence of H_2_/N_2_ flow conditions, these compounds undergo a continuous transformation, producing NH_3_ over an extended period of approximately 7 days, sustaining a catalytic cycle with an impressive activity of up to 2.8 mmol·g^−1^·h^−1^. Although the reaction mechanism was not explicitly detailed, it was speculated that the MvK mechanism might be involved, as the N_2_ dissociation step occurred rapidly and was not the RDS.

Wang et al.’s [241] discovery identifies early 5f-element catalysts, particularly the surfaces of ThO_2_ and UO_2_ (111), as exceptional and highly efficient single-component catalysts for the conversion of N_2_ to NH_3_. Their pioneering study highlights atomic oxygen vacancies as the central and crucial active sites for NH_3_ synthesis from N_2_ and H_2_ gases. Their analysis shows that the chemically adsorbed N≡N molecule is highly resistant to direct dissociation, with significant energy barriers of 3.64 eV and 2.91 eV on ThO_2_ and UO_2_ surfaces, respectively. This suggests that NH_3_ synthesis is unlikely to proceed via the dissociative LH mechanism.

While N_2_ binds to the vacancy site on these catalysts, H_2_ dissociates directly at the same location, producing two H^−^ ions. These H^−^ ions then migrate across the surface, enabling the hydrogenation of the distal N site on adsorbed N_2_, which ultimately leads to the formation of N=NH and N=NH_2_ intermediates. The N-N bond cleavage occurs through the reaction N-NH_2_ + H → NH + NH_2_, although this reaction step does not precisely mirror the step from Figure 7g to Figure 7h in which the latter involves the formation of an N-NH_3_ intermediate. The subsequent steps (depicted in Figure 7i–m) for the generation of the second NH_3_ molecule, however, proceed identically. The most energetically favorable and computed reaction pathway is as follows: N_2_(g) → [N≡N] → [N=NH] → [N-NH_2_] → [HN∙∙∙NH_2_] → [HN∙∙∙NH_3_] → [NH] → [NH_2_] → [NH_3_] → 2NH_3_(g). The process may follow an associative MvK mechanism.

The nickel-loaded lanthanum nitride (Ni/LaN) catalyst exemplifies the MvK mechanism [53]. The rate-limiting step is the fusion of hydrogen with the nitrogen lattice (N + H → NH), with an energy barrier of ~0.54 eV, closely matching the Arrhenius plot value (~0.622 eV). Further hydrogenation selectively targets the top nitrogen of adsorbed N≡N at the vacancy site, initiating a stepwise progression toward the formation of a second NH_3_. This sequence, culminating in the release of NH_3_, is governed by an overall energy barrier of 0.76 eV.

The account by Tian et al. [242] provides additional details on other catalysts, e.g., Ru/CaFH, BaH_2_–BaO/Fe, Ba–Co@NC, (Co, Fe)/BaCeO_3–*x*_N*_y_*H*_z_* ATiO_3–*x*_H*_x_* (A = Ba, Sr, Ca) and Ru-loaded BaTiO_2.5_H_0.5_ [243], where the MvK mechanism is at play.

### 5.5. Associative Alternating (Symmetric) Pathway (AAP)

The associative alternating (symmetric) pathway (AAP) [201] involves the adsorption of H_2_ on the catalyst’s surface, its dissociation, and the alternating hydrogenation of adsorbed N_2_. N_2_ binds end-on orientation to the catalyst, which weakens the N≡N bond. Dissociatively adsorbed hydrogen atoms are then progressively transferred to N_2_ through an alternating hydrogenation process, ultimately producing NH_3_.

The steps of the AAP mechanism are illustrated in Figure 8a–m. Starting from the reference state (a), the mechanism includes the clean catalyst (a), non-dissociatively adsorbed N_2_ (b), and dissociatively adsorbed H_2_ (c, f, and i). The surface reaction between H and the distal N atom in N_2_ forms intermediate hydrogenated species N=NH (d). Sequential addition of H atoms to the N atoms alternately, in a symmetric fashion, leads to the formation of NH-NH (e), NH-NH_2_ (g), N_2_H_2_ (h), and NH_2_NH_2_ (j). The cleavage of the N-N bond occurs during the hydrogenation stages (i → j), followed by the desorption of the first NH_3_ at stage (j → k), leaving an amine group (-NH_2_) at stage (k). Further hydrogenation leads to the formation of the second NH_3_ (l), followed by its desorption from (l) to (m).

The elementary steps, a-n, of the AAP mechanism are depicted in Figure 8a–m. The process begins from the reference state (a), which includes the clean catalyst surface, a molecule of N_2_, and three molecules of H_2_, represented as [] + N_2_(g) + 3H_2_(g). In step (b), N_2_ adsorbs non-dissociatively onto the surface, typically in an end-on configuration. This is followed by the dissociative adsorption of the first H_2_ molecule in step (c), providing two surface-bound hydrogen atoms. One hydrogen then reacts with the distal nitrogen of N_2_ to form the N–NH intermediate (d). Subsequently, symmetric addition of hydrogen atoms leads to the formation of NH–NH (e), followed by the dissociative adsorption of the second H_2_ molecule in step (f), enabling further hydrogenation to produce NH–NH_2_ (g) and then NH_2_–NH_2_ (h). The third H_2_ molecule dissociates in step (i), leading to the formation of the NH_2_···NH_3_ intermediate in step (j). The cleavage of the N–N bond and the desorption of the first NH_3_ molecule occur during the transition from step (j) to (k), leaving behind an –NH_2_ group. Continued hydrogenation results in the formation of the second NH_3_ molecule in step (l), which desorbs in step (m). Final desorption of both NH_3_ molecules is completed in step (n), thereby regenerating the clean catalyst surface and closing the catalytic cycle.
a.[] + N_2_(g) + 3H_2_(g)Reference state: clean catalyst and unreacted N_2_ and H_2_b.[] + N_2_(g) ⇌ [N_2_]Molecular end-on adsorption of N_2_(g)c.[N_2_] + H_2_(g) ⇌ [N-N + H H]Dissociative adsorption of first H_2_ moleculed.[N-N + H H] ⇌ [N-NH + H]Formation of N-NH intermediatee.[N-NH + H] ⇌ [NH-NH]Formation of NH-NH intermediatef.[NH-NH] + H_2_(g) ⇌ [NH-NH + H H]Dissociative adsorption of second H_2_ moleculeg.[NH-NH + H H] ⇌ [NH-NH_2_ + H]Formation of NH-NH_2_ intermediateh.[NH-NH_2_ + H] ⇌ [NH_2_-NH_2_]Formation of NH_2_-NH_2_ intermediatei.[NH_2_-NH_2_] + H_2_(g) ⇌ [NH_2_-NH_2_ + H H]Dissociative adsorption of third H_2_ moleculej.[NH_2_-NH_2_ + H H] ⇌ [NH_2_-NH_3_ + H]Formation of NH_2_∙∙∙NH_3_ intermediatek.[NH_2_-NH_3_ + H] ⇌ [NH_2_ + H] + NH_3_(g)Formation of NH_2_ and desorption of first NH_3_l.[NH_2_ + H] + NH_3_(g) ⇌ [NH_3_]Adsorption of second NH_3_(g)m.[NH_3_] ⇌ [] + NH_3_(g)Desorption of second NH_3_(g)n.[] + 2NH_3_ ⇌ []Desorption of two NH_3_ molecules, leaving behind the clean catalyst

In the vacancy-assisted AAP mechanism, schematically represented in the study by Yet al. [203], the first H_2_ directly hydrogenates end-on adsorbed N_2_ in an alternating fashion, forming NH=NH via the LR mechanism. The next H_2_ dissociates on the catalyst surface, with one hydrogen atom adsorbing while the other hydrogenates NH=NH, yielding the NH-NH_2_ intermediate. Further hydrogenation produces NH_2_-NH_2_, and the third H_2_ follows a similar pattern, leading to the release of the first NH_2_. Subsequent hydrogenation releases a second NH_2_ molecule. This reaction mechanism closely resembles that shown in Figure 8, differing only in how H_2_ participates in the hydrogenation process.

A recent study on the catalyst Ca_3_CrN_3_H(001) [108] examined the AAP mechanism, demonstrating that the free energy pathway for N_2_ hydrogenation follows the alternating route. N_2_ activation and hydrogenation primarily occur at Ca cations rather than Cr sites. The N_2_ + H → N=NH step is the most energy-intensive, requiring 0.63 eV when hydrogen is supplied from lattice H. The process becomes even less favorable with gaseous H_2_, raising the energy cost to 0.85 eV. Further hydrogenation proceeds either through direct gaseous H_2_ attack on the N moiety or by filling a surface H vacancy, followed by lattice H attack. Preferential hydrogenation occurs at the surface-bound N rather than the distal N, making the alternating associative pathway approximately 0.5 eV more favorable. Partial cleavage of NH=NH and NH-NH_2_ bonds may occur during the process, while NH_2_-NH_2_ bond dissociation is believed to happen spontaneously. Upon further hydrogenation, NH_2_ forms as an intermediate, and final hydrogenation leads to the exothermic release of NH_2_ (−1.55 eV).

Li et al. [244] reported an FeN_3_-embedded graphene single-atom catalyst (SAC) using DFT modeling, demonstrating that the FeN_3_ active site significantly facilitates N≡N bond cleavage, making both distal and alternating pathways of the associative mechanism highly favorable. Similarly, Liu et al. [156] proposed that Fe_3_ clusters anchored on θ-Al_2_O_3_(Fe_3_/θ-Al_2_O_3_(010)) exhibit an extraordinary turnover frequency, outperforming Fe’s C_7_ sites by two orders of magnitude. This remarkable enhancement is attributed to the unparalleled ability of Fe_3_ clusters to catalyze N_2_ activation via the AAP mechanism, where adsorbed N_2_ first hydrogenates to form an N=NH intermediate. This pathway is more efficient than the dissociative mechanism involving direct N_2_ cleavage.

### 5.6. Associative (Asymmetric) Distal Pathway (ADP)

The associative (asymmetric) distal pathway (ADP) [68,245,246] involves the hydrogenation of molecularly adsorbed N_2_ on transition metal surfaces, where the terminal nitrogen atom plays the first involved in facilitating the reaction. In this process, hydrogen molecules first dissociate on the catalyst surface, releasing activated (surface-bound) hydrogen atoms that are transferred to N_2_.

The catalytic reduction of nitrogen via the distal pathway proceeds through a sequential hydrogenation mechanism, as illustrated in Figure 9a–m. The process begins with the clean catalyst surface exposed to gaseous nitrogen and hydrogen (a). N_2_ adsorbs molecularly in an end-on configuration onto the active site (b), followed by dissociative adsorption of the first H_2_ molecule, leading to surface-bound hydrogen atoms (c). These hydrogen atoms facilitate the stepwise hydrogenation of the distal nitrogen atom, producing N–NH and then N–NH_2_ intermediates (d–e). A second H_2_ molecule adsorbs and dissociates (f), allowing further hydrogenation to form N–NH_3_ (g). At this stage, the first NH_3_ molecule is released, leaving behind the [N H] intermediate on the surface (h). This remaining nitrogen atom undergoes additional hydrogenation, leaving behind the N-H intermediate (i). The third H_2_ molecule dissociates (j), and subsequent steps convert the N–H species to NH_2_ and then to NH_3_ (k–l). The second ammonia molecule is then desorbed from the surface (m), regenerating the catalyst and completing the cycle with the release of both NH_3_ molecules (see step n).
(a)[] + N_2_(g) + 3H_2_(g)Reference state: clean catalyst, unreacted N_2_ and H_2_(b)[] + N_2_(g) ⇌ [N_2_]Molecular end-on adsorption of N_2_(g)(c)[N_2_] + H_2_(g) ⇌ [N-N + H H]Dissociative adsorption of first H_2_ molecule(d)[N-N + H H] ⇌ [N-NH + H]Formation of N-NH intermediate(e)[N-NH + H] ⇌ [N-NH_2_]Formation of N-NH_2_ intermediate(f)[N-NH_2_] + H_2_(g) ⇌ [N-NH_2_ + H H]Dissociative adsorption of second H_2_ molecule(g)[N-NH_2_ + H] ⇌ [N-NH_3_ + H]Formation of N-NH_3_ intermediate(h)[N-NH_3_ + H] ⇌ [N H] + NH_3_(g)Formation of NH_2_-NH_2_ and desorption of first NH_3_(i)[N-H] ⇌ [N-H]Formation of NH intermediate(j)[N-H] + H_2_(g) ⇌ [N-H + H H]Dissociative adsorption of third H_2_ molecule(k)[N-H + H H] ⇌ [NH_2_ + H]Formation of second NH_2_(l)[NH_2_ + H] ⇌ [NH_3_]Adsorption of second NH_3_(g)(m)[NH_3_] ⇌ [] + NH_3_(g)Desorption of second NH_3_(g)(n)[] + 2NH_3_(g) ⇌ []Desorption of both NH_3_ molecules

Catalysts like Ca_3_CrN_3_H(001) [108] effectively promote the distal pathway by forming lattice hydrides and weakly adsorbing H_2_, which helps prevent hydrogen poisoning. The catalyst’s adsorption sites are approximately 3 Å apart—an ideal distance for hydrogenation. However, stable dissociative adsorption of N_2_ requires a separation of about 7.20 Å, corresponding to the spacing between equivalent Ca_3_ active sites. This configuration forces the bottom nitrogen atom deeper into the Ca_3_ cavity toward Cr, resulting in high-energy adsorption states. Consequently, dissociative N_2_ adsorption at these sites demands extensive surface diffusion, making this pathway both spatially and energetically unfavorable.

While less thermodynamically favorable than the alternating pathway, the ADP pathway may still occur on the catalyst, especially when N_2_ is strongly adsorbed or when catalyst properties—such as steric and electronic factors—favor distal hydrogenation. After the release of the first NH_3_ molecule, it was suggested that the remaining nitrogen at a Cr site in Ca_3_CrN_3_H undergoes hydrogenation to form NH and NH_2_, which weakens the Cr^−^N coordination and leads to the formation of NH_3_. The ammonia desorption process is endothermic by approximately 1 eV.

The study of Liu et al. [68] identified an associative distal mechanism within a ferrierite catalyst with dual Mo(II) sites, [L-2Mo^(II)^/FER], which promotes efficient ammonia synthesis under ambient conditions. With a significantly lower energy barrier for N_2_ hydrogenation (0.73 eV) compared to direct N≡N bond breaking (1.38 eV), the catalyst outperforms Ru in turnover frequency over a wide temperature range, highlighting the potential of tailored porous catalysts with multiple active sites for the Haber–Bosch process.

The pure siliceous zeolite-supported Ru SAC (Ru SAs/S-1) catalyst is another exemplary system [107] that facilitates the ADP mechanism. DFT calculations indicate that H_2_ weakly adsorbs on Ru sites or lattice oxygen sites within the zeolite framework. Notably, H_2_ molecules remain largely intact within the zeolite channels, where they are physically adsorbed with a weak interaction energy of 0.21 eV. By contrast, N_2_ preferentially adsorbs linearly on Ru with a stronger binding energy of 0.57 eV.

The reaction step N_2_ + H_2_ → N-NH_2_ (the first hydrogenation step), distinct from that shown in Figure 9d, was proposed as the rate-determining step, with an activation energy of 0.88 eV. This is reasonable because the subsequent reaction steps, N-NH_2_ + H_2_ → NH + NH_3_↑, NH + H_2_ → NH_3_↑, have lower activation energies, facilitating their faster progression. Depicted in Figure 9a′–j′ is the distal pathway, similar to that used in the study by Qui et al. [107]. It includes the following reaction steps: (a′) [M]; (b′) [M-N_2_] + 3H_2_(g); (c′) [M-N_2_ H_2_] + 2H_2_(g); (d′) [M-N-NH_2_] + 2H_2_(g); (e′) [M-N-NH_2_ H_2_] + H_2_(g); (f′) [M-NH∙∙∙NH_3_] + H_2_(g); (g′) [M-NH] + NH_3_(g) + H_2_(g); (h′) [M-NH H_2_] + NH_3_(g); (i′) [M-NH_3_] + NH_3_(g); (j′) [M] + 2NH_3_(g). In each step of the reaction, Figure 9a′–j′, the total number of hydrogen and nitrogen moieties is conserved, although this stoichiometric balance is not explicitly depicted in the schematic. In this case, the physically adsorbed H_2_ molecule directly reacts with the adsorbed nitrogen species to form NH_3_, which then desorbs. The reaction pathway is more consistent with an ER-type ADP mechanism.

Wang et al. [247] have designed homogeneous single-atom Ru centers on an H-ZMS-5 (HZ) support, with Ru atoms individually anchored in the micropores of HZ. Their DFT calculations revealed that the direct dissociation of N_2_ into two nitrogen atoms on a single Ru site of the Ru/HZ SAC catalyst requires a substantial energy barrier of 2.90 eV. However, the hydrogenation process via the ADP mechanism, leading to the formation of N^−^NH_2_, demands a significantly lower activation energy of just 1.12 eV. The hydrogenation of N_2_ was crucial for the significant weakening of the N≡N bond, with the RDS proposed to be associated with forming the N_2_H_2_ intermediate.

Ghuman et al. [248] suggested that using an associative mechanism, N_2_H (N_2_ + H → N_2_H) serves as the RDS for both Ru/MgO and RuFe/MgO catalysts. However, while this pathway applies to the former catalyst, it could shift to N_2_H_2_ + H → N_2_H_3_ in the latter case, as this transition involves higher energy. For the dissociative mechanism, a significant energy barrier was observed for the reaction NH_2_ + H → NH_3_ on both catalysts, but the associative pathway was found favorable over the dissociative one.

Bai and colleagues [117] proposed that the rate-determining step in their associative pathway is N_2_ + H → N + NH on the Ru/CeO_2_ catalyst. They demonstrated N_2_H as the precursor of the N≡N bond cleavage. However, the lack of a detailed reaction pathway in their study makes it difficult to discern whether a distal path or an alternating one was followed prior to the dissociation of the kinetically relevant vertically adsorbed N=NH intermediate, in contrast to the widely known dissociative route [5,249,250]. According to the schematic model presented by the authors, the N_2_ molecule initially aligns in an end-on orientation on the clean catalyst. Upon protonation of the distal nitrogen, the molecule shifts to a side-on orientation before the N=NH bond undergoes cleavage. The subsequent hydrogenation of the adsorbed NH and N species results in the formation of NH_3_.

### 5.7. The Associative Enzymatic Reaction Pathway (AERP)

In the associative enzymatic reaction pathway (AERP), employed by nitrogenase enzymes in biological systems, N_2_ is converted to NH_3_ through stepwise hydrogenation, and the catalytic site is subsequently regenerated. This mechanism, which avoids the high-energy nitrogen dissociation step, is increasingly used as a model for photocatalytic and electrocatalytic NRR [251].

A recent mechanism proposed by Hau et al. [252] resembles AERP in its alternating hydrogenation pattern and is similar to the associative alternating pathway (AAP, see Figure 8). This “bridge distal pathway” (BDA) involves side-on (μ-η^2^:η^2^) adsorption of N_2_, followed by sequential hydrogenation and eventual release of two NH_3_ molecules. The “bridge” term refers to this specific binding mode, distinguishing BDA from AAP where N_2_ is not necessarily bridge-bound. However, it remains uncertain whether the hydrogen atoms involved in reduction are sourced from the catalyst surface or the gas phase. In a consecutive-type pathway [162], three hydrogen atoms attack one nitrogen atom in pre-adsorbed N_2_, leading to the stepwise release of NH_3_, as proposed by Hou et al. [252].

Figure 10 illustrates the sequential steps of AERP, where two ammonia molecules are synthesized from a single N_2_ molecule. Similar to BDA but unlike AAP, N_2_ adopts a side-on orientation, enhancing hydrogenation efficiency. The orientation of N_2_ influences its accessibility and reactivity, with the side-on configuration in AERP creating a more favorable environment for N≡N bond cleavage. Ultimately, the preferred reaction pathway depends on the catalyst’s surface morphology and electron density, both of which govern N_2_ activation and reactivity.

The sequential steps of AERP are outlined below (steps a–m). Since the intermediate steps of hydrogenation mirror those of the AAP with the end-on orientation of N_2_ (Figure 8), a detailed reiteration of each step is unnecessary.
a.[] + N_2_(g) + 3H_2_(g)Reference state: clean catalyst unreacted N_2_ and H_2_b.[] + N_2_(g) ⇌ [N_2_]Molecular side-on adsorption of N_2_(g)c.[N_2_] + H_2_(g) ⇌ [N-N + H H]Dissociative adsorption of first H_2_ moleculed.[N-N + H H] ⇌ [N-NH + H]Formation of N-NH intermediatee.[N-NH + H] ⇌ [NH-NH]Formation of NH-NH intermediatef.[NH-NH] + H_2_(g) ⇌ [NH-NH + H H]Dissociative adsorption of second H_2_ moleculeg.[NH-NH + H H] ⇌ [NH-NH_2_ + H]Formation of NH-NH_2_ intermediateh.[NH-NH_2_ + H] ⇌ [NH_2_-NH_2_]Formation of NH_2_-NH_2_ intermediatei.[NH_2_-NH_2_] + H_2_(g) ⇌ [NH_2_-NH_2_ + H H]Dissociative adsorption of third H_2_ moleculej.[NH_2_-NH_2_ + H H] ⇌ [NH_2_-NH_3_ + H]Formation of NH_2_∙∙∙NH_3_ intermediatek.[NH_2_-NH_3_ + H] ⇌ [NH_2_ + H] + NH_3_(g)Formation of NH_2_ and desorption of first NH_3_l.[NH_2_ + H] + NH_3_(g) ⇌ [NH_3_]Adsorption of second NH_3_(g)m.[NH_3_] ⇌ [] + NH_3_(g)Desorption of second NH_3_(g)n.[] + 2NH_3_ ⇌ []Desorption of two NH_3_, leaving the clean catalyst

Peng et al. investigated Ru-Co dual single-atom catalysts (RuCo DSAC) for ammonia synthesis via AERP [253]. They identified that the first hydrogenation of N_2_, forming N_2_H (N_2_ + H → N_2_H) on the Ru site, had the highest kinetic barrier, making it the rate-limiting step. Their findings are noteworthy, showing that the NH_3_ synthesis rate of RuCo DSAC was 8.2 times and 7.5 times higher than that of Co SAC and Ru SAC, respectively, at 400 °C. A combination of experimental data and DFT calculations revealed that the synergistic interaction between Ru and Co centers significantly enhanced NH_3_ synthesis via the associative pathway.

Similarly, theoretical investigations of the singly dispersed bimetallic catalyst Rh_1_Co_3_/CoO(011) [192] demonstrated alternating hydrogenation of N_2_, with H_2_ activation occurring on both metal sites. Other studies have explored different catalytic systems, such as Pt_2_@C_3_N_3_ and Ru_3_@C_3_N_3_, focusing on the AERP mechanism, as discussed by others [254,255].

Zhang et al. [113] examined Ru clusters on Sm_2_O_3_ (Ru/Sm_2_O_3_) and found that both dissociative and associative pathways contribute to ammonia synthesis. They proposed two models for side-on adsorbed N_2_: Model I (hydrogen-lean) and Model II (hydrogen-rich surface). In both models, N_2_ adsorption was endothermic (*E*_ad_ = 0.29 eV for Model I, 0.25 eV for Model II), while dissociative adsorption was exothermic (*E*_ad_ = −0.34 eV for Model I, −0.42 eV for Model II). Protonation proceeded via an associative pathway, but it did not follow enzymatic, distal, or alternating mechanisms, ultimately yielding two NH_3_ molecules.

In Model I, the reaction followed this sequence:

N_2_ → 2N → 2N + 2H → NH + N + H → NH + NH → NH + NH + 2H → NH_2_ + NH + H → NH_2_ + NH + 2H → NH_3_ + NH + H → NH + H → NH_2_ + H → NH_3_(g). For the associative pathway, the process remained endothermic until the breakdown of the N=N-H intermediate: N_2_ → N_2_ + H_2_ → N=N-H + H → N + NH + H. They found that the dissociation of N=NH into N and NH ad-species was both kinetically and thermodynamically favorable under the reaction conditions. The subsequent steps followed a similar pattern to the associative portion of the pathway.

### 5.8. Mechanism of Nitrogenase-Catalyzed Ammonia Synthesis and Associative Analogues

While this overview focuses primarily on the Haber–Bosch process, it should be appreciated that nitrogen fixation occurs through three main pathways: (i) natural phenomena such as lightning, (ii) biological conversion by nitrogenase enzymes found in specific microorganisms, and (iii) industrial synthesis via the Haber–Bosch process [256].

Below, we briefly outline the widely accepted mechanism by which nitrogenase catalyzes ammonia synthesis. In the AERP, hydrogen atoms typically originate from molecular hydrogen (H_2_) present in the medium. By contrast, enzymatic systems may also supply protons (H^+^) or hydride ions (H^−^), interacting with water molecules through protonation or deprotonation steps, as commonly observed in acid–base catalysis. While AERP represents a biologically inspired mechanism, related pathways such as the distal and alternating mechanisms are also known to be relevant under various catalytic conditions [256,257].

Nitrogenase operates under ambient conditions by coupling electron transfer with ATP hydrolysis [256,258], where ATP refers adenosine triphosphate. The enzyme’s active site binds N_2_ in a suitable orientation for multi-step reduction while receiving electrons from a reductase partner. These enzymes enable plants to assimilate nitrogen, making it accessible throughout the food chain. Nitrogenases are primarily found in bacteria and archaea, such as *Rhizobium* in legumes and *Azotobacter* in soils.

Three types of nitrogenases are known: molybdenum-dependent (Mo-Nase), vanadium-dependent (V-Nase), and iron-only (Fe-Nase) [259]. Each enzyme contains a complex metallocluster: the [Fe_8_S_7_] P-cluster, which serves as an electron relay, and the active-site cofactor, [MFe_7_S_9_C] (where M = Mo, V, or Fe), where N_2_ reduction takes place [260]. Electrons required for the reaction are donated by a reductase protein (NifH) containing a [Fe_4_S_4_] active site. This reductase is itself reduced by ferredoxin or flavodoxin.

The nitrogenase-catalyzed reaction is energetically demanding due to the high N≡N bond dissociation enthalpy of 941.4 kJ mol^−1^ [44]. Breaking this bond electrochemically would require an overpotential of approximately −1.6 V—well beyond the limits of the nature’s “electrochemical window”, i.e., in an aqueous environment at pH 7 this is limited by the evolution of O_2_ (+0.82 V) at one end and the evolution of H_2_ (−0.41 V) at the other end. Nitrogenase overcomes this challenge by coupling electron transfer to ATP hydrolysis, allowing the reaction to proceed under mild conditions. The mechanism of nitrogenase is complex (see [261] and references therein); what is generally accepted has been summarized recently [262] but many questions remain [263], and involves multiple electron transfer steps.

The nitrogenase cycle requires eight electrons and protons, and its stoichiometry is given by Equation (2), where ADP and Pi refer adenosine diphosphate and inorganic phosphates, respectively.N_2_ + 8*e*^−^ + 10H^+^ + 16ATP → 2 NH_4_^+^ + H_2_ + 16ADP + 16P_i_(2)

This equation emphasizes several crucial features of the nitrogenase mechanism, such as the role of ATP hydrolysis in facilitating substrate reduction and the mandatory production of 1 mole of H_2_ for every mole of N_2_ reduced. This byproduct of hydrogen is essentially a waste of two reducing equivalents and four ATP molecules for each molecule of N_2_ reduced.

Electrons are delivered by dinitrogenase reductase (NifH), which contains a [Fe_4_S_4_] cluster and is itself reduced by a ferredoxin or flavodoxin. Upon ATP binding, NifH forms a complex with the catalytic component, enabling electron transfer from the [Fe_4_S_4_] site to the P-cluster, and ultimately to the active FeMo cofactor. ATP hydrolysis plays a key role, providing the thermodynamic driving force for the reaction by lowering the redox potential needed for electron transfer, effectively bypassing nature’s electrochemical constraints (see [260,264] and references therein). The complex dissociates after each electron transfer, with phosphate release constituting the rate-limiting step. The entire process proceeds through eight one-electron transfer steps, designated E_0_ to E_7_, as outlined in the Lowe–Thorneley model [265] (Figure 11). The first NH_3_ molecule is released after the third hydrogenation, and the second after the fifth, with the release of H_2_ occurring at various stages. The exact pathway can follow either a distal or alternating mechanism, depending on the sequence of hydrogenation steps and the intermediates formed.

Nitrogen binding to the cofactor typically occurs after the accumulation of three to four electrons, often accompanied by the release of H_2_. N_2_ is then progressively reduced via a diazene-type intermediate to form two NH_3_ molecules. However, the mechanism is not fully resolved. Unproductive H_2_ loss from E_2_(2H), E_3_(3H), or E_4_(4H) states can reverse progress by two electrons per event, adding complexity to the overall cycle.

The electrons transferred to the FeMo cofactor may reduce the metal centers or form bridging hydrides [266,267,268,269]. If hydride species are present, they are expected to emerge after two-electron accumulation steps, while odd-numbered E-states may feature reduced metal centers. Upon H_2_ release, the cofactor reaches a highly reduced state, enabling effective N_2_ activation and subsequent conversion to ammonia.

While the initial steps of nitrogen fixation up to the formation of the Janus E_4_ intermediate are relatively well established, the exact sequence of events in the latter half of the mechanism remains an open question [256,270,271]. Two primary pathways have been proposed: the distal and alternating mechanisms. In the distal pathway, hydrogenation is directed first toward the nitrogen atom furthest from the metal center, which is fully reduced and released as ammonia before the proximal nitrogen is hydrogenated. By contrast, the alternating pathway involves a sequential, back-and-forth addition of hydrogen atoms to both nitrogen atoms—alternating between the distal and proximal sites—until both are converted to ammonia.

**Figure 11 ijms-26-04670-f011:**
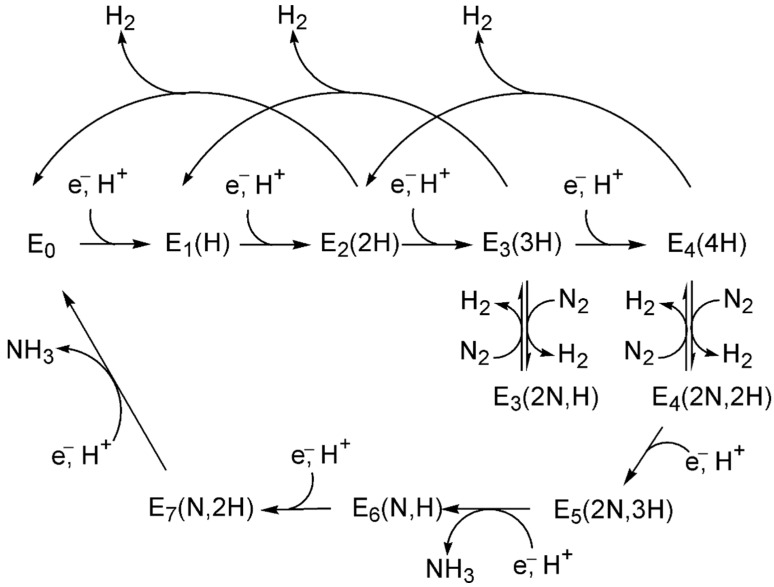
The Lowe–Thorneley mechanism of nitrogenase (adapted from [262,263,272]), emphasizing the coupled transfer of electrons and protons across eight steps. While the complete model indicates that N_2_ binds at either the E3 or E4 stages, this representation focuses less on the pathway through E3. Additionally, the model includes protons that are associated with the FeMo cofactor.

## 6. Reaction Order and the Rate-Determining Step

The reaction order is a key and informative metric reflecting the RDS in a catalytic process [24,75,102,106]. It provides critical insight into the reaction mechanism [118]. The reaction orders for N_2_, H_2_, and NH_3_ may be represented by α, β, and γ [114]. The ammonia synthesis rate under the Haber–Bosch process’s commercial conditions, as Temkin and Pyzhev proposed in 1940 [32,273], is related to α. Across all catalysts, the reaction exhibits nearly first-order kinetics with respect to N_2_ and close to zero order in H_2_ and NH_3_ [116,274]. However, the reaction order with respect to H_2_ varies depending on the chosen promoter [77], as evidenced by the changes in reaction order observed for H_2_ passing over unpromoted, Cs-promoted, and Ba-Cs-promoted Ru catalysts. Others have shown that the introduction of Ba and La promoters effectively reduces the H_2_-induced inhibition observed in Cs-promoted Ru/MgO catalysts [274].

The signs of α, β, and γ, whether positive or negative, reveal the nature of the molecular interactions with the catalyst surface. A positive reaction order sometimes suggests weak adsorption, meaning the molecule’s physical coverage on the catalyst surface is minimal, while a negative order indicates strong adsorption [70], which may obstruct the adsorption of other species. For instance, the negative β values of −0.14, −0.18, and −0.76 were reported for H_2_ on Ru/MgO, Ru/CeO_2__500red, and Cs^+^/Ru/MgO_500red, respectively, implying strong adsorption of hydrogen on the Ru surface, which impedes the efficient activation of molecular N_2_ [114]. This phenomenon, known as hydrogen poisoning, limits the preferential activity of Ru-based catalysts [76,275], such as Ru/MgO and Cs-Ru/MgO [27,116,274].

By contrast, the positive reaction orders of 0.50, 0.43, 0.32 1.5, 1.2, and 0.8 were reported for H_2_ on RuFe/MgO [248], Co/Ba/La_2_O_3__700red, Co/La_2_O_3__700red [114], BaH_2_-BaO/Fe/CaH_2_ [122], Co-Ba/C, and Co_3_Mo_3_N [118], respectively. The β values range from 0.1 to 1.7 for catalysts like Fe/SrNH, Co/SrNH, Ni/SrNH, and Ni/CaNH [116], meaning that these catalysts are not subject to hydrogen poisoning and demonstrate an increased ammonia synthesis rate [116,276].

γ is typically negative with respect to NH_3_. A highly negative γ may indicate that an increased concentration of ammonia on the catalyst slows down the reaction rate, in accordance with le Chatelier’s principle. The catalyst Co/La_2_O_3__700red showed a γ of −0.51, pointing to the inhibitory effect of adsorbed NH, NH_2_, and NH_3_ species, which prevents the NH_3_ yield from reaching equilibrium [114]. However, adding Ba to Co/La_2_O_3__700red reduced this negative value to −0.17, indicating that Ba promotes the desorption of the adsorbate, accelerating the reaction even as it approaches equilibrium.

Fe-based catalysts (e.g., Fe/K(3)/MgO-500red and Fe/MgO-500red) exhibit significantly larger negative γ values compared to Ru-based catalysts (Cs^+^/Ru/MgO-500red and Ru/CeO_2_-500red) [70]. This behavior is attributed to the strong adsorption of NH_x_ species on the Fe surface, where the equilibrium between NH_x_ and NH_3_ inhibits N_2_ adsorption and activation. Drummond et al. [77] observed that γ for the doubly promoted Ru catalyst could shift from −0.17 to −0.59 with a 40 °C increase, indicating greater sensitivity of the reaction rate to ammonia concentration at higher temperatures. This shift suggests the possibility of back reactions, site blocking, or a change in catalyst behavior not observed in unpromoted or Cs-promoted Ru catalysts.

In conventional heterogeneous catalysis, the α value for N_2_ typically hovers near unity (0.8–1.0) [118]. Specific examples include Co NPs (α = 1.08) [277], Ru/MgO (α = 1.4), RuFe/MgO (α = 0.97) [248], Fe/K(3)/MgO-500red (α = 1.0), Ru/CeO_2_-500red (α = 0.85) [70], Cs^+^/Ru/MgO-500red (α = 1.07), Ru/CeO_2_-500red (α = 0.85), Co/Ba/La_2_O_3_-700red (α = 0.85), Co/La_2_O_3_-700red (α = 0.97) [114], commercial Fe (α = 0.90) [27], Ru-Cs/MgO (α = 0.99) [121], and Ru/[Ca_24_Al_28_O6_4_]^4+^(O_2_^−^)_2_ (α = 0.75–1.0) [75]. This near-unity order suggests that the RDS in the slow reaction process is the dissociation of molecular N_2_ [56,70,277].

In electride-based catalysts, however, N_2_ dissociation is no longer the RDS [24,75,102]. Electrically conductive electrides efficiently donate electrons to the metal surface (e.g., Ru nanoparticles), enhancing N_2_ cleavage and accelerating the synthesis of NH_3_. As a result, hydrogenation becomes the rate-limiting step, reducing α for N_2_ from 0.5 to ~1.5 and lowering the apparent activation energy to 0.45–0.65 eV, thereby improving the efficiency of ammonia synthesis. Such reaction orders were reported for Ca-based systems such as Ru/Ca(NH_2_)_2_ (α = 0.53), Ru/Ca_2_N:*e*^−^ (α = 0.53), Ru/C_12_A_7_:*e*^−^ (α = 0.46), Ru/BaO-CaH_2_ (α = 0.47), Ru/CaH_2_ (α = 0.57) [106] and Ru/[Ca_24_Al_28_O_64_]^4+^(O_2_^−^)_2−x_ (*e*^−^)_2x_ (α = 0.57) [75].

For the CeNi_2_ and CeNi_5_ catalysts, α values of 0.663 and 0.884 were reported [127], respectively, suggesting that the RDS involves a hydrogenation step for NH_x_ in the case of CeNi_2_, while N_2_ dissociation is the RDS for CeNi_5_. This accords with the observation that CeNi_2_ reinforces higher ammonia synthesis activity.

For Ni/ReN (Re = Ce, La) catalysts, α was estimated to be 1.2 [27]. In both cases, however, the RDS for ammonia synthesis is the hydrogenation of lattice nitrogen via the reaction N + H → NH, occurring on the surface of LaN and CeN, respectively. For Ni/YN, however, the overall activation barrier is primarily governed by the hydrogenation of NH_2_ species. For CeN NPs, Co_3_Mo_3_N, and Ru/Ba-Ca(NH_2_)_2_, the value of α was 1.0 [27], despite the activation energy being relatively low (Table 1).

The reaction orders for the Ni/CeN catalyst were determined to be α (N_2_) = 1.2 (1.2), β (H_2_) = 1.6 (1.2), and γ (NH_3_) = −1.4 (−1.7) [27]. For CeNi_2_, the corresponding values were 0.663, 1.089, and −0.725, while for CeNi_5_, they were 0.884, 0.633, and −0.690, respectively [127]. These β values revealed the absence of hydrogen poisoning issues on these catalysts. The negative value of γ for all three catalysts indicated notable adsorption of NH_3_, leading to coverage on the catalyst surface. For the catalysts, including Ni/CeN, CeN, Ni/LaN, Co_3_Mo_3_N, LaRuSi, Ru/Ba-Ca(NH_2_), and Cs-Ru/MgO, and BaH_2_-BaO/Fe/CaH_2_, Ru/BaH_2_-BaO, Ru powder, and Ru-Cs/MgO, the reaction orders with respect to NH_3_ were approximately −1.4, −1.6, −1.7, −1.3, −1.05, −0.9, −0.35 [27], −1.1 and −1.7 [122], −0.15 and −0.12 [118], respectively.

The reaction order of N_2_ over the intermetallic catalyst LaCoSi was reported as 0.45 [106], suggesting that N_2_ dissociation occurs rapidly enough to maintain a surface saturated with activated nitrogen. This behavior closely resembles that of high-performance ruthenium-loaded electride catalysts [118] and Co–LiH [131], where the RDS is no longer N_2_ cleavage. Kinetic analysis, involving linear regression between calculated and experimental synthesis rates [106], assessed reaction rates over LaCoSi using rate equations derived from various reaction steps. The analysis was based on three key assumptions: (1) ammonia synthesis proceeds via eight elementary steps, (2) one step controls the overall reaction rate, and (3) adsorption follows the Langmuir model.

This analysis yielded a relatively low *R*^2^ value of 0.583 for N_2_ activation, whereas hydrogenation steps, N + H → NH, NH + H → NH_2_, and NH_2_ + H → NH_3_, had significantly higher *R*^2^ values of 0.930, 0.982, and 0.987, respectively. The highest correlation for hydrogenation steps strongly suggests that the RDS for ammonia synthesis over LaCoSi is one of the NH_x_ formation steps rather than N_2_ dissociation. This conclusion aligns with findings for ruthenium-loaded C_12_A_7_:*e*^−^, where a similar shift in the RDS was observed [118]. Morimoto et al. [278] reported a similar trend for Ru and Ru + K catalysts but suggested that temperature dependence might determine whether the RDS involves N + H → NH or NH_2_ + H → NH_3_.

For Co/SrNH, Fe/SrNH, Ni/SrNH, and Ni/CaNH catalysts, the N_2_ reaction order falls within a narrow range of 1.1–1.2, while H_2_ exhibits a broader range from 1.7 (Co/SrNH) and 1.6 (Fe/SrNH) to significantly lower values of 0.2 (Ni/CaNH) and 0.1 (Ni/SrNH) [116]. The drastic reduction in β for Ni-based catalysts is attributed to slow consumption of dissociatively adsorbed H, leading to hydrogen accumulation on the Ni surface and resistance to hydrogen poisoning. Higher activation energies—0.973 eV for Ni/CaNH and 0.955 eV for Ni/SrNH, compared to 0.546 eV for Co/SrNH and 0.492 eV for Fe/SrNH—further support this distinction, suggesting a unique catalytic mechanism for Ni/CaNH and Ni/SrNH.

A dual reaction mechanism [116] has been proposed for hydrogenation on Ni-based catalysts, involving N_2_ activation at (1) Co metal sites and (2) NH_2_^−^ vacancy sites on the SrNH support. The RDS for ammonia formation is the coupling of H with NH_2_^−^ vacancies (NH_2_ + H → NH_3_), a process observed in Ni/SrNH and exhibiting a high activation barrier, similar to Co/SrNH, Ni/LaN, and Ni/CeN. Among these, Co/SrNH demonstrates the highest catalytic efficiency for ammonia synthesis.

For Ru/Ba(10)-TiH_2_ and Ru–Cs/MgO catalysts [76], the respective values of α, β, and γ were 0.15 (−0.59), 0.79 (0.89), and −0.36 (0.11). These results indicate that Ru/Ba(10)-TiH_2_ could exhibit appreciable ammonia synthesis activity and is less susceptible to hydrogen poisoning than Ru–Cs/MgO.

Potassium modifies the ammonia reaction order from −0.6 to −0.35 and the hydrogen reaction order from 0.76 to 0.44 on the (100) and (111) faces of iron [72]. However, within experimental error, the activation energy remains unchanged, implying that the fundamental ammonia synthesis mechanism is not altered. The increase in apparent ammonia order from −0.6 on clean Fe(100) to −0.35 on K/Fe(100) reflects this modification.

For Ru(0001), the RDS is proposed to be N_2_ dissociation, with an activation energy of 1.3 eV, despite DFT calculations suggesting that NH_2_ formation presents the highest energy barrier [146]. Tautermann et al. [279] noted that at low temperatures (<200 K), the reactions N + H → NH and NH + H → NH_2_ may act as the RDS, though this is irrelevant to the Haber–Bosch process. Zhang et al. [144] further corroborated this, reporting activation energies of 1.29 eV for N + H → NH and 1.36 eV for N_2_ → 2N, suggesting that N_2_ dissociation is the true rate-limiting step.

## 7. Discussion and Conclusions

Jocobi [280] highlighted discrepancies between real-world catalyst performance and single-crystal DFT predictions, emphasizing the importance of studying 2 nm Ru metal particles on various supports to advance understanding of ammonia synthesis over Ru-based catalysts. This discrepancy arises because typical single-crystal DFT models represent idealized, flat surfaces—such as Ru(0001)—that fail to capture the diversity of active sites present on real nanoparticle catalysts. In practice, Ru and Fe NPs may expose a mix of terraces, steps, edges, ledges, kinks, and corners—features that generate undercoordinated sites like B5 and C7, which often serve as the true reactive centers for N_2_ dissociation. Small slab models (e.g., 2 × 2 unit-cells with ~36 atoms) are too limited to include such sites. To capture them, one must construct larger supercells (e.g., 4 × 3 or 5 × 3) or explicitly model high-index facets like Ru(10$\bar{1}$2) or stepped surfaces such as Ru(11$\bar{2}$1) and Ru(10$\bar{1}$5), which naturally expose these low-coordination geometries. These models, typically exceeding 100 atoms, are computationally demanding, but necessary for bridging the gap between theoretical predictions and actual catalytic behavior, particularly in reactions like N_2_ activation that are sensitive to surface structure.

Beyond B5 and C7 sites, step-edge, corner (e.g., intersection of multiple facets on an NP), and defect sites significantly influence N_2_ dissociation. These sites, especially at step-edges, kinks or corners, are more reactive than terrace sites due to their lower coordination number. For instance, corner sites—where two steps meet—are particularly reactive, as are defect-oriented sites like vacancies, which stabilize reactive intermediates. These surface features—though their mechanistic roles are yet to be fully elucidated—are crucial for enhancing catalytic efficiency by promoting the activation and cleavage of the N≡N bond, followed by subsequent hydrogenation and desorption steps.

There exists no singular, universally applicable rule for the straightforward identification of active sites in newly designed or yet-to-be-identified catalysts that facilitate the molecular or dissociative adsorption of N_2_ without thorough theoretical inspection. Consequently, the mechanistic pathways governing ammonia synthesis activity processes cannot be generalized across all catalytic systems. Nevertheless, the associative mechanism emerges as predominant over its dissociative counterpart when the adsorption strength of molecular N_2_ exceeds a critical threshold around −1.0 eV. The manifestation of a particular reaction mechanism is intricately orientation-dependent for N_2_, intertwined with the nature of the chemical or physical interactions governing reactant adsorption. Furthermore, hybrid mechanistic pathways, characterized by associative and dissociative components, may dominate under certain catalytic conditions, particularly in vacancy-assisted catalytic systems, wherein enhanced mechanistic complexity is likely to arise.

While different catalysts follow varying reaction mechanism, such as LH for Ru, Co, and Fe, ER/MvK for transition metal nitrides, or distal, alternating, and mixed associative-dissociative concerted pathways for ammonia synthesis, the exploration of these mechanisms remains confined to a limited set of catalyst systems. If molecular adsorption of N_2_ is stronger than its dissociative adsorption, the LH mechanism may not be applicable, as it depends on the dissociation of N_2_ into reactive intermediates (such as atomic N). In such cases, alternative mechanisms such as the LR, ER, MVK, or hybrid models could be more suitable, where N_2_ remains molecularly adsorbed, and hydrogen atoms or H_2_ participate in the reaction through different pathways. Therefore, to optimize catalyst design, researchers should focus on elucidating all possible pathways for each catalyst using advanced experimental techniques, including isotope labeling, surface science methods, and in-situ spectroscopy, alongside computational modeling. These approaches will yield a deeper understanding of the intricate processes occurring on the catalyst surface. Furthermore, developing high-throughput screening techniques for evaluating diverse catalysts will enhance our ability to design more efficient and stable transition metal heterogeneous catalysts for ammonia synthesis.

The use of promoters enhances ammonia synthesis activity through two distinct mechanisms. First, they induce an electronic charge transfer to the catalyst, generating active sites that strengthen the nitrogen-iron bond through moderate adsorption while weakening the nitrogen-nitrogen bond. This dual action activates the adsorbed N_2_ and facilitates its seamless dissociation. Second, they orchestrate localized surface modifications, creating specialized sites where NH_3_, produced during hydrogenation, is weakly adsorbed. This process not only eases the desorption of NH_3_ but also prevents surface blockage, ensuring the continuous and efficient adsorption of N_2_, thereby facilitating the recycling of the catalytic process.

While many studies have proposed dissociatively adsorbed N_2_ [60] and the desorption of NH_3_ [36] from the catalyst surface, such as Mo_2_N [281,282] and Ru [15] as potential rate-limiting steps, these are consistent with the suggestions that have been made [283,284] based on the theory of Temkin and Pyzhev [273]. Recent discussions, however, support the idea proposed by Enomoto et al. [285] in 1952, which suggests that the reaction N + H → NH (and other hydrogenation steps) could be the slow step in the process [31] and, therefore, should be considered as the RDS.

For most Ru-based catalysts, the RDS is the dissociation of N_2_, indicated by near-unity α values, suggesting N_2_ activation as the primary rate-limiting factor. However, in ruthenium-loaded electrides, the RDS shifts to the hydrogenation step. This shift occurs because the electride phase provides a highly efficient electron source, facilitating faster N_2_ cleavage and lowering the energy required for N_2_ activation. As a result, the hydrogenation step becomes the new rate-limiting step, lowering the α value for N_2_ (α < 1) and reducing the overall activation energy. The shift in RDS is further influenced by the specific electronic structure of the electrides, which alters the interactions between hydrogen and the catalyst surface, making the hydrogenation step more pronounced. However, for catalysts such as Co/C12A7:*e*^−^, Ni/CeN, Ni/LaN, and CeN NPs, and Co_3_Mo_3_N [24], which feature low activation energies (*E*_a_ << 60 kJ mol^−1^), α remains close to 1.0, even though the RDS is the hydrogenation step (N + H_x_ → NH_x_) due to a complex interplay of vacancy-driven hydrogenation.

Variations in β reflect the impact of hydrogen adsorption, with hydrogen poisoning observed in some Ru-based catalysts, which can be mitigated by promoters such as Ba, Ce and La (and their oxides), among others. These promoters modify the catalyst surface topology, enhancing charge transfer and reducing the negative effects of hydrogen poisoning by altering hydrogen adsorption energies. Despite these insights, the relationship between RDS and reaction order remains incompletely understood, particularly regarding the specific conditions—such as catalyst composition, temperature, and reaction environment—that trigger the shift in RDS.

Future research should perhaps focus on elucidating the atomic and electronic mechanisms through which promoters influence hydrogen adsorption and the RDS while investigating temperature and pressure dependencies across diverse catalyst systems. These insights will be crucial for predicting catalyst performance under industrial conditions. However, these mechanistic advances must be considered within the broader context of the challenges faced by ammonia synthesis, particularly regarding energy consumption and carbon emissions.

Despite over a century of dominance in industrial ammonia synthesis, the Haber–Bosch process remains an energy-intensive and carbon-emitting technology, largely due to its reliance on high-temperature, high-pressure conditions and hydrogen derived from fossil fuels. Advances in catalyst design, integration with renewable hydrogen (enabling green ammonia), and emerging electrochemical methods have shown promise, but remain limited by cost, scalability, and technical maturity. From a policy perspective, accelerating the adoption of low-carbon alternatives will require coordinated investment in R&D, infrastructure for green hydrogen, and regulatory frameworks that support sustainable production without jeopardizing global food security. Looking ahead, novel catalyst architectures—such as electrides, nitrides, hydrides, and bi-, tri-, and multi-metallic (high-entropy) alloy systems—may offer pathways to improved efficiency. Ultimately, the long-term goal is to emulate nature’s own solution: nitrogenase enzymes, which fix nitrogen at ambient temperature and pressure with remarkable selectivity and efficiency.

## Figures and Tables

**Figure 1 ijms-26-04670-f001:**
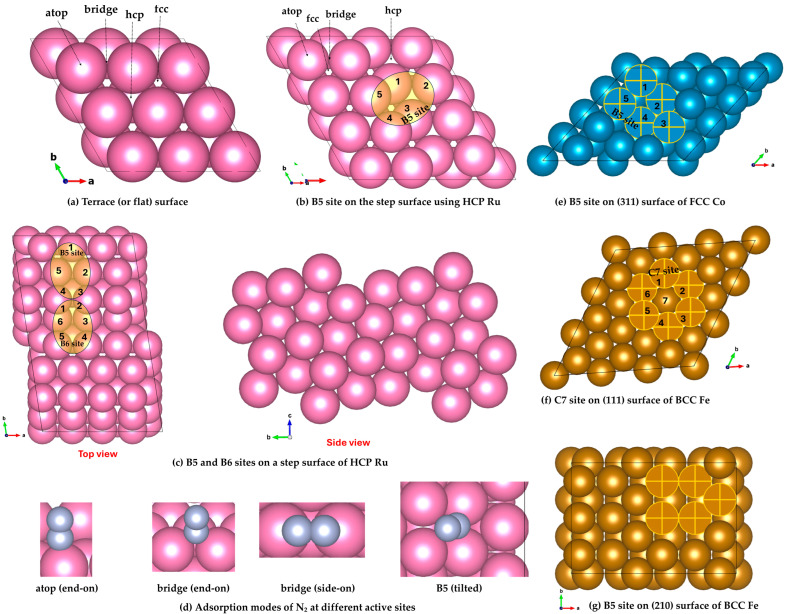
(**a**,**b**) Illustration (top view) of common adsorption sites on the flat/step HCP(0001) surface of Ru (space group: *P6_3_/mmc*), including on-top (directly above a surface atom), bridge (between two adjacent atoms), face-centered cubic (fcc) hollow, and hexagonal close-packed (hcp) hollow sites. The fcc site is a threefold hollow site where the adsorbate sits above a triangle of atoms stacked in an fcc-like sequence, typically found on (111) surfaces of fcc metals. The hcp site is also a threefold hollow, but lies above atoms following an hcp stacking, characteristic of hcp metals like Ru. (b) Manual construction of a monoatomic step surface on Ru(0001) by selectively removing two rows of atoms from the top layer; this approach does not involve the use of a Wulff construction. (**b**,**c**) The B5 site is a step-edge site on a stepped Ru surface, composed of five metal atoms—two from the step and three from the terrace—within an hcp lattice. This site is well-known for its catalytic role in ammonia synthesis. (**c**) Marking of the B5 and B6 sites on a step surface of HCP Ru; the slab is shown from two different perspectives. (**d**) Schematic illustrations of different adsorption modes of molecular N_2_ on various surface sites of Ru. Illustrations of the B5, C7, and B6 adsorption sites on the (311), (111), and (210) surfaces of FCC Co and BCC Fe are shown in panels (**e**), (**f**), and (**g**), respectively. Atom colors: Ru—pink; Co—deep cyan; Fe—deep orange; N—light blue.

**Figure 2 ijms-26-04670-f002:**
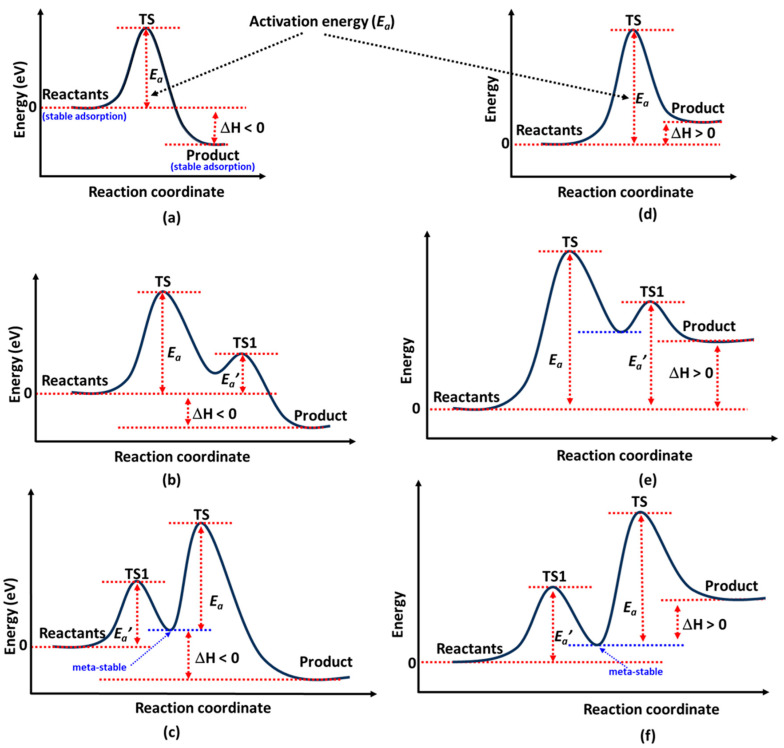
(**a**–**f**) Schematic representation of the energy of activation as a function of reaction coordinate. TS and TS1 are transition states, Δ*H* is the reaction enthalpy, and *E*_a_ and *E*_a_*’* are the activation energies. The label at ‘0’ energy corresponds to the reference point. The peaks representing *E*_a_ correspond to the actual transition states.

**Figure 3 ijms-26-04670-f003:**
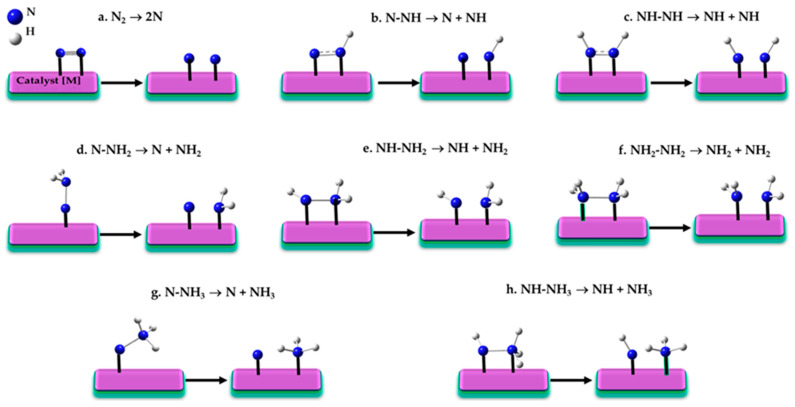
A schematic illustration of the possible dissociative pairs likely involved in the entire nitrogen reduction reaction (NRR) pathway on heterogeneous transition metal catalysts. These intermediates are central to NRR for Haber–Bosch catalysts and are likely to be relevant for photocatalytic and electrocatalytic processes [171].

**Figure 4 ijms-26-04670-f004:**
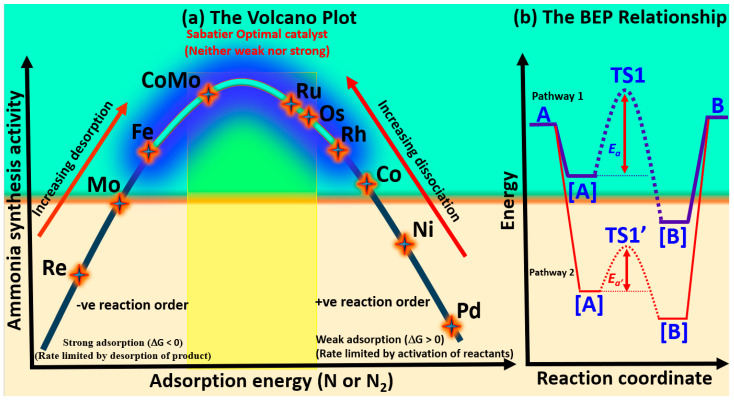
(**a**) Schematic representation of the widely used volcano plot illustrating the limitations of transition metal catalysts in ammonia synthesis. Δ*G* refers to the Gibbs free energy. Shown in (**b**) is the representation of the Brønsted–Evans–Polanyi (BEP) relationship (see text for description).

**Figure 5 ijms-26-04670-f005:**
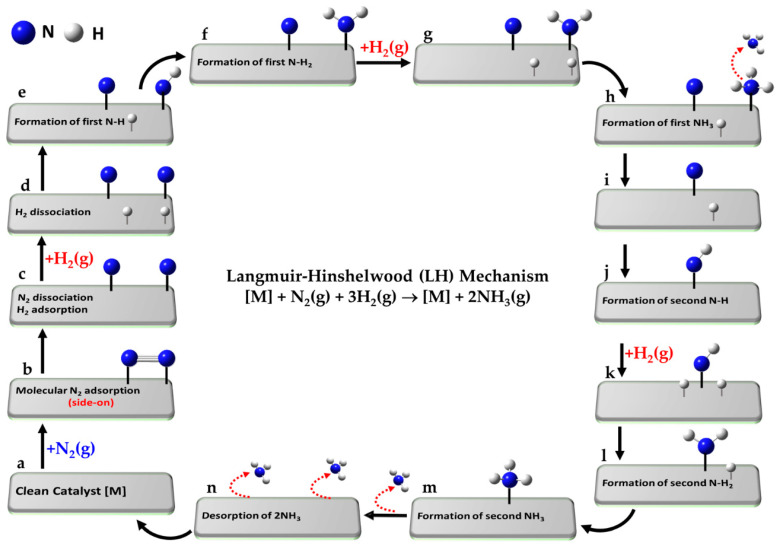
(**a**–**n**) Schematic representation of the Langmuir–Hinshelwood (LH) (dissociative) mechanism for ammonia synthesis. The elementary reaction is [M] + N_2_(g) + 3H_2_(g) → [M] + 2NH_3_(g), where M in [M] represents the active site on the clean catalyst [] and (g) denotes the gas phase species.

**Figure 6 ijms-26-04670-f006:**
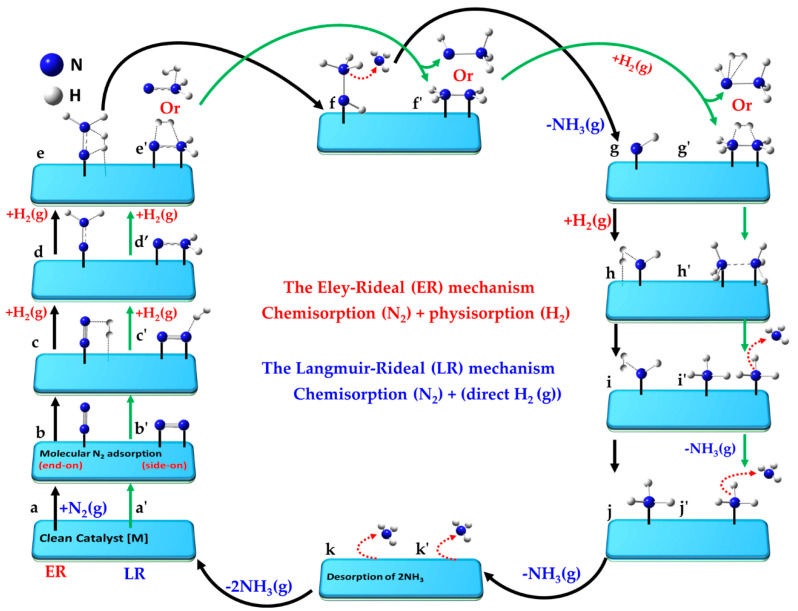
Schematic representation of the (**a**–**k**) (associative) Eley–Rideal (ER) (**left**) and (**a’**–**k’**) Langmuir–Rideal (LR) (**right**) mechanisms for ammonia synthesis. The elementary reaction, [M] + N_2_(g) + 3H_2_(g) → [M] + 2NH_3_(g), is shown, where M in [M] represents the active site on the clean catalyst []. Two possible adsorption modes of molecular N_2_ on the catalyst are depicted, facilitating direct chemical reactions between chemisorbed N_2_ and physisorbed (or gas-phase) H_2_ molecules via a non-dissociative mechanism, leading to the formation of intermediate and product species along the ER and LR pathways, respectively. In both cases, the N_2_ molecule may be adsorbed in either an end-on or side-on orientation. Interaction modes between the reactants are illustrated in (**f′**,**g′**), each showing the varied nature of plausible intermediate species.

**Figure 7 ijms-26-04670-f007:**
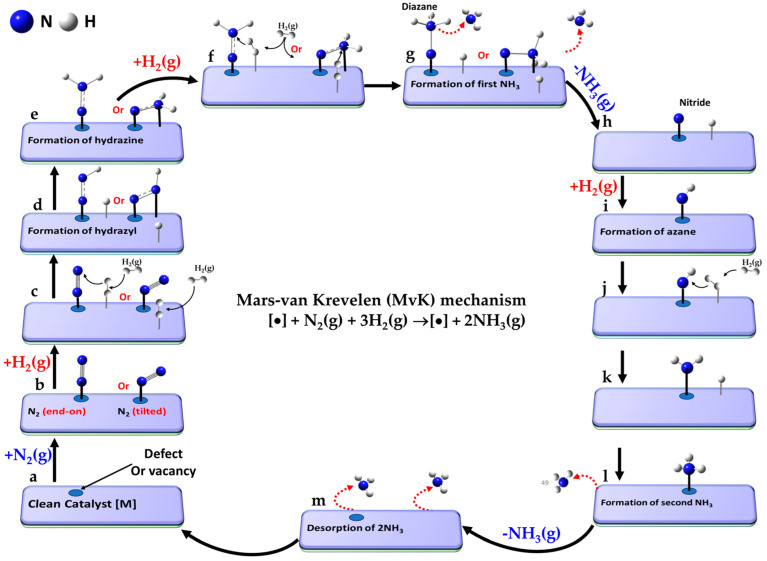
(**a**–**m**) Schematic representation of the associative Mars–van Krevelen (MvK) mechanism for ammonia synthesis, following a (distal-type) consecutive pathway for hydrogenation. The elementary reaction is as follows: [•] + N_2_(g) + 3H_2_(g) → [•] + 2NH_3_(g), where [•] represents the vacancy site on the clean catalyst. Two possible modes of adsorption of N_2_ are shown: (**left**) end-on; (**right**) (side-on tilted).

**Figure 8 ijms-26-04670-f008:**
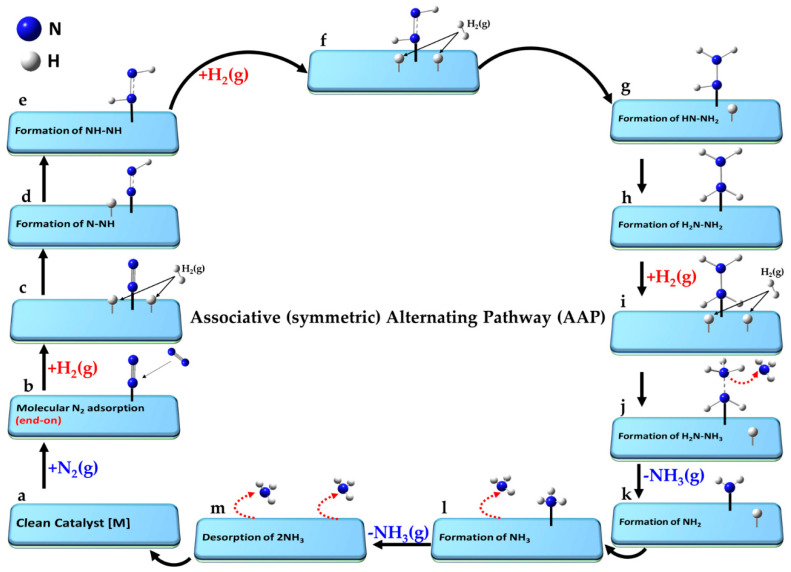
(**a**–**m**) Schematic representation of the associative alternating (symmetric) pathway for ammonia synthesis. The elementary reaction is as follows: [M] + N_2_(g) + 3H_2_(g) → [M] + 2NH_3_(g), where M in [M] represents the active site on the clean catalyst [] (e.g., Ca_3_CrN_3_H [108]).

**Figure 9 ijms-26-04670-f009:**
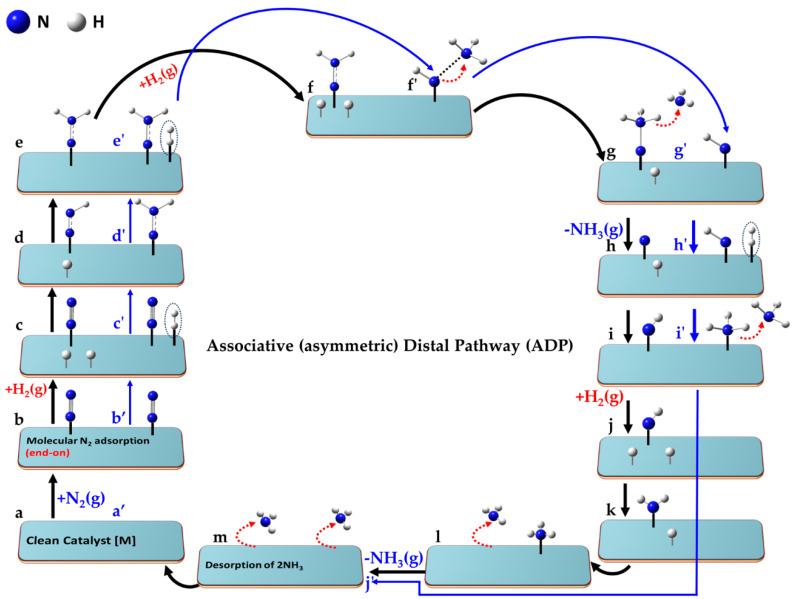
(**a**–**m**) Schematic representation of the associative (asymmetric) distal pathway (ADP) mechanism for ammonia synthesis. The elementary reaction is as follows: [M] + N_2_(g) + 3H_2_(g) → [M] + 2NH_3_(g), where M in [M] represents the active site in the clean catalyst [] and (g) denotes the gas phase species. Shown in (**a′**–**j′**) is the distal pathway similar to that utilized in the study of Qui et al. [107], including the following reactant, intermediate and product steps: (**a′**) [M] + N_2_(g) + 3H_2_(g); (**b′**) [M-N_2_] + 3H_2_(g); (**c′**) [M-N_2_ H_2_] + 2H_2_(g); (**d′**) [M-N-NH_2_] + 2H_2_(g); (**e′**) [M-N-NH_2_ H_2_] + H_2_(g); (**f′**) [M-NH∙∙∙NH_3_] + H_2_(g); (**g′**) [M-NH] + NH_3_(g) + H_2_(g); (**h′**) [M-NH H_2_] + NH_3_(g); (**i′**) [M-NH_3_] + NH_3_(g); (**j′**) [M] + 2NH_3_(g). In each step of the reaction, (**a′**–**j′**), the total number of hydrogen and nitrogen moieties is conserved, although this stoichiometric balance is not explicitly depicted in the schematic.

**Figure 10 ijms-26-04670-f010:**
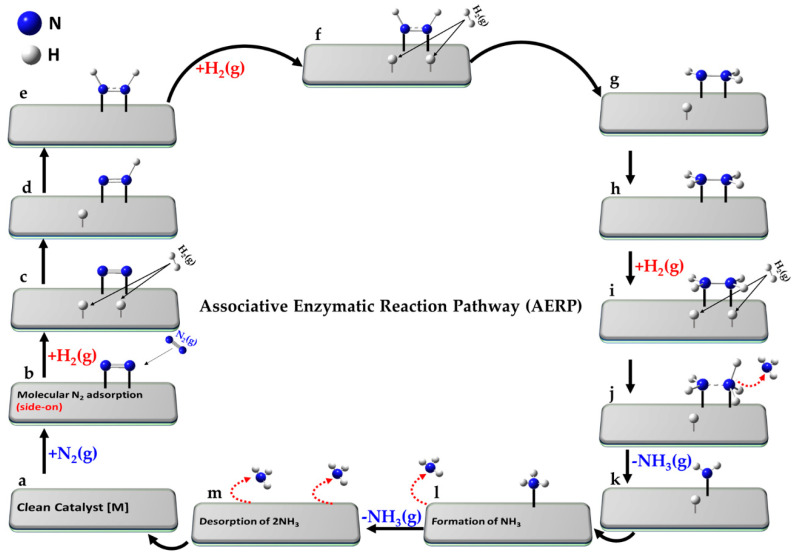
(**a**–**m**) Schematic representation of the enzymatic reaction pathway (AERP) for ammonia synthesis. The elementary reaction is as follows: [M] + N_2_(g) + 3H_2_(g) → [M] + 2NH_3_(g), where [M] represents the clean catalyst and (g) denotes the gas phase species.

## Data Availability

This research used data reported in the manuscript itself.

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
