# Peer review of "Ammonia Synthesis over Transition Metal Catalysts: Reaction Mechanisms, Rate-Determining Steps, and Challenges"

_ijms, 2025, doi:10.3390/ijms26104670_

Round 1
Reviewer 1 Report
Comments and Suggestions for Authors
Comments and Suggestions for Authors
In this review article, the authors discuss the ammonia synthesis process, examining potential reaction mechanisms, the activation energy and the rate-determining step of the reaction that may occur with various transition metal catalysts. The manuscript is quite extensive, containing a wealth of information from the research published in both earlier and more recent scientific writings. Schemes of elementary reaction mechanisms are also provided to enhance comprehension of the complex processes of adsorption, surface reaction, and desorption that take place in heterogeneous catalysis.
I would like to share the following insights.
- I suggest correcting this typographical mistake in the keywords: The Harbor-Bosch Method
- I propose that electrons (e), which appear in equation (2) and Figure 1, be denoted by the symbol e¯ (having a negative charge).
- These two sentences could be presented more clearly for readers; perhaps a succinct explanation of the concept of nature's electrochemical window would also be relevant.
-on page 2, lines 72-74 „(in an aqueous environment, potentials between the oxidation of H2O with evolution of O2 and the reduction of H+ to produce H2; under standard conditions at pH 7 this is +817 mV and –413 mV, respectively.)”
- on page 2, lines 83-84 „It is the hydrolysis of ATP that lowers the redox potential for the transfer of the electron from the P-cluster to the cofactor, the work-around Nature’s electrochemical window.”
- Any figures sourced from other articles should include a copyright acknowledgment in their captions.
5. I recommend that in Table 1, the references be listed individually in the final column of the table.
Author Response
Reply to reviewer 1
============
In this review article, the authors discuss the ammonia synthesis process, examining potential reaction mechanisms, the activation energy and the rate-determining step of the reaction that may occur with various transition metal catalysts. The manuscript is quite extensive, containing a wealth of information from the research published in both earlier and more recent scientific writings. Schemes of elementary reaction mechanisms are also provided to enhance comprehension of the complex processes of adsorption, surface reaction, and desorption that take place in heterogeneous catalysis.
Reply: We sincerely thank the reviewer for their positive and constructive feedback. We are pleased that the manuscript’s comprehensive coverage of ammonia synthesis, including the discussion of potential reaction mechanisms, activation energies, and rate-determining steps, was well appreciated. We also appreciate the recognition of our efforts to include schemes that help clarify the complex processes involved in heterogeneous catalysis. Your comments encourage us to further enhance the clarity and depth of our analysis, and we are grateful for your thoughtful evaluation.
I would like to share the following insights.
- I suggest correcting this typographical mistake in the keywords: The Harbor-Bosch Method
Reply: We have correct this.
- I propose that electrons (e), which appear in equation (2) and Figure 1, be denoted by the symbol e¯ (having a negative charge).
Reply: We renamed Fig. 1 as Fig. 11 (pg. 34), and corrected e as e¯, as suggested.
- These two sentences could be presented more clearly for readers; perhaps a succinct explanation of the concept of nature's electrochemical window would also be relevant.
-on page 2, lines 72-74 „(in an aqueous environment, potentials between the oxidation of H2O with evolution of O2 and the reduction of H+ to produce H2; under standard conditions at pH 7 this is +817 mV and –413 mV, respectively.)”
- on page 2, lines 83-84 „It is the hydrolysis of ATP that lowers the redox potential for the transfer of the electron from the P-cluster to the cofactor, the work-around Nature’s electrochemical window.”
Reply: We have addressed the reviewer’s concerns in Section "5.8 Mechanism of Nitrogenase-Catalyzed Ammonia Synthesis and Associative Analogues," where we provide a detailed clarification.
- Any figures sourced from other articles should include a copyright acknowledgment in their captions.
Reply: Most of the figures presented in this work are original drawings created by us. Unless otherwise stated, we have appropriately cited any copyrighted materials referenced in the manuscript.
- I recommend that in Table 1, the references be listed individually in the final column of the table.
Reply: We revised the table as suggested.
Reviewer 2 Report
Comments and Suggestions for Authors
The review article by Varadwaj et al. presents a thorough and insightful summary of ammonia synthesis, focusing on nitrogen reduction mechanisms and the rate-determining steps associated with various transition metal catalysts in heterogeneous systems. The authors commence by outlining the significance of the renowned Haber-Bosch process, highlighting the inherent challenges in dinitrogen reduction, and subsequently comparing this with nitrogenase enzyme-mediated ammonia synthesis.
They proceed to discuss activation energies, citing reported literature values and illustrating energy profiles corresponding to various transition states. The article then delves into the elementary reaction mechanisms, offering a critical analysis supported by volcano plot evaluations. Detailed discussions are provided on the dissociative Langmuir-Hinshelwood (LH) mechanism and the associative Langmuir-Rideal (LR) mechanism. Furthermore, the associative Mars-van Krevelen (MvK) mechanism is comprehensively explained and systematically compared with the LH and LR pathways.
The alternating associative pathway (AAP) and alternating distal pathway (ADP) are also incorporated and contextualized effectively within the broader discussion. These mechanisms are further compared with the enzymatic pathway, namely the alternating enzymatic reduction pathway (AERP). The review concludes with an examination of reaction orders and rate-determining steps (RDS), offering valuable insights and well-supported conclusions.
The authors have conducted a comprehensive review of the relevant literature, and the manuscript is well-written, clear, and appropriately cited. The emphasis on heterogeneous dinitrogen reduction and its associated mechanism is suitably positioned and effectively underscores its significance, thereby capturing the attention of both academic and industrial communities. In light of this, I recommend the publication of this paper in International Journal of Molecular Sciences (Int. J. Mol. Sci.), contingent upon minor revisions.
- The title should reflect that these mechanisms are considered for heterogeneous ammonia synthesis. A general title could imply both, but the authors have focused solely on heterogeneous transition metal catalysts.
- I would appreciate the inclusion of a figure illustrating the B5-type sites (line 169), hcp site (line 276), fcc site, and bridge (line 277), as the meaning of these terms may not be immediately clear to the reader.
-. Figure 3: It would be beneficial to label the figures wherever possible, using terms such as ‘‘top-on’’, ‘‘end-on’’, ‘‘side-on’’, etc., to enhance clarity.
- Line 322: "N-NH → N-NH" (Figure 3b) should be corrected to "N-NH → N + NH."
- Line 466: A space is needed between "Nâ‚‚ dissociation."
- Line 480: Remove one of the brackets in “[200, 201]”
- Line 483: ‘‘…a form of weak [200]’’ – A form of weak what? This statement is unclear and requires clarification
- Line 666: Remove one period after “[140, 216]” as there are two periods.
- Line 1042: What does "single crystal DFT prediction" refer to?
Author Response
Reply to reviewer 2
---------------
The review article by Varadwaj et al. presents a thorough and insightful summary of ammonia synthesis, focusing on nitrogen reduction mechanisms and the rate-determining steps associated with various transition metal catalysts in heterogeneous systems. The authors commence by outlining the significance of the renowned Haber-Bosch process, highlighting the inherent challenges in dinitrogen reduction, and subsequently comparing this with nitrogenase enzyme-mediated ammonia synthesis.
They proceed to discuss activation energies, citing reported literature values and illustrating energy profiles corresponding to various transition states. The article then delves into the elementary reaction mechanisms, offering a critical analysis supported by volcano plot evaluations. Detailed discussions are provided on the dissociative Langmuir-Hinshelwood (LH) mechanism and the associative Langmuir-Rideal (LR) mechanism. Furthermore, the associative Mars-van Krevelen (MvK) mechanism is comprehensively explained and systematically compared with the LH and LR pathways.
The alternating associative pathway (AAP) and alternating distal pathway (ADP) are also incorporated and contextualized effectively within the broader discussion. These mechanisms are further compared with the enzymatic pathway, namely the alternating enzymatic reduction pathway (AERP). The review concludes with an examination of reaction orders and rate-determining steps (RDS), offering valuable insights and well-supported conclusions.
The authors have conducted a comprehensive review of the relevant literature, and the manuscript is well-written, clear, and appropriately cited. The emphasis on heterogeneous dinitrogen reduction and its associated mechanism is suitably positioned and effectively underscores its significance, thereby capturing the attention of both academic and industrial communities. In light of this, I recommend the publication of this paper in International Journal of Molecular Sciences (Int. J. Mol. Sci.), contingent upon minor revisions.
Reply: We sincerely thank the reviewer for the thoughtful and encouraging assessment of our manuscript. We are pleased that the overall structure, clarity, and depth of our analysis—particularly on nitrogen reduction mechanisms and rate-determining steps—were well received. We appreciate the recognition of our comparative treatment of the dissociative and associative pathways, as well as the contextual inclusion of the enzymatic and Mars–van Krevelen mechanisms. We have addressed the minor revisions as suggested and are grateful for the reviewer’s recommendation for publication in International Journal of Molecular Sciences.
- The title should reflect that these mechanisms are considered for heterogeneous ammonia synthesis. A general title could imply both, but the authors have focused solely on heterogeneous transition metal catalysts.
Reply: We have revised the title to better reflect the scope of the discussion. While the primary focus is on heterogeneous transition metal catalysts, the elementary mechanisms examined are also applicable to other catalytic systems. We believe the updated title more accurately captures this broader relevance.
- I would appreciate the inclusion of a figure illustrating the B5-type sites (line 169), hcp site (line 276), fcc site, and bridge (line 277), as the meaning of these terms may not be immediately clear to the reader.
-. Figure 3: It would be beneficial to label the figures wherever possible, using terms such as ‘‘top-on’’, ‘‘end-on’’, ‘‘side-on’’, etc., to enhance clarity.
Reply: We appreciate the reviewer’s suggestion regarding the clarity of adsorption site terminology. In response, we have added a new section titled “2. Catalytic Surface Reactivity Through the Lens of Geometry: Adsorption Site Topologies and Their Role in Heterogeneous Catalysis”, which explicitly addresses the B5-type, HCP, FCC, and bridge sites discussed in the text. Additionally, we have included clear labels such as “top-on”, “end-on”, and “side-on” to improve visual understanding of adsorption sites.
- Line 322: "N-NH → N-NH" (Figure 3b) should be corrected to "N-NH → N + NH."
Reply: We have corrected the concern as follows: N-NH → N + NH
- Line 466: A space is needed between "Nâ‚‚ dissociation."
Reply: We have corrected this.
- Line 480: Remove one of the brackets in “[200, 201]”
Reply: We have corrected this.
- Line 483: ‘‘…a form of weak [200]’’ – A form of weak what? This statement is unclear and requires clarification
Reply: We have corrected this, and we apologize for the typographical error.
- Line 666: Remove one period after “[140, 216]” as there are two periods.
Reply: We have corrected this.
- Line 1042: What does "single crystal DFT prediction" refer to?
Reply: In computational catalysis, "single-crystal DFT models" typically refer to slab models of low-index, well-ordered surfaces (such as Ru(0001), Fe(110), or Co(0001)) that mimic the surface of a bulk crystal. These slabs are periodic in two dimensions and often constructed with 3–5 atomic layers and a relatively small surface unit cell (e.g., 2×2 or 3×3). While these models are computationally efficient and capture the basic adsorption behavior on flat terraces, they often fail to represent the structural diversity and local environments found on real catalyst particles.
For example, B5 sites — known to be highly active for Nâ‚‚ dissociation in ammonia synthesis over Ru — are step-edge configurations involving five metal atoms: two from an upper step and three from the terrace. These configurations do not exist on flat (0001) surfaces and cannot be captured in small unit cells that only model ideal terraces. To include a B5 site, one needs to construct larger supercells (e.g., a 4×2 or 5×3 surface) or explicitly model high-index facets like Ru(10Ì…1Ì…2) or stepped surfaces such as Ru(11Ì…21), which naturally expose such low-coordinated step-edge geometries.
As an estimate, while a typical 3×3 Ru(0001) slab with ~4 layers contains ~36 atoms and is tractable with standard DFT, modeling a stepped surface that includes a B5 site may require >100 atoms, especially if multiple layers are needed to converge surface energies and adsorption geometries. This increases computational cost by at least an order of magnitude, especially when exploring multiple adsorption configurations, transition states, or coverage effects.
Therefore, while single-crystal models provide valuable insights into fundamental surface chemistry, they often overlook the most catalytically relevant active sites. Accurately capturing these undercoordinated motifs — such as B5, C7, or kinked configurations — requires larger, high-index, or nanoparticle models, which are computationally demanding but essential for realistic mechanistic understanding. We have included a summary of this point in the Discussion and Conclusion section for better clarity and to address the reviewer’s concern.